# Mechanistic basis of antimicrobial resistance mediated by the phosphoethanolamine transferase MCR-1

Allen P. Zinkle[1,8], Mariana Bunoro Batista[2,8], Carmen M. Herrera [3], Satchal K. Erramilli [4], Brian Kloss [1], Khuram U. Ashraf [1], Kamil Nosol[4], Guozhi Zhang[5], Rosemary J. Cater[1,6], Michael T. Marty[5], Anthony A. Kossiakoff [4], M. Stephen Trent [3] ✉, Rie Nygaard [1,7] ✉, Phillip J. Stansfeld [2] ✉ & Filippo Mancia [1] ✉

Polymyxins are used to treat infections caused by multidrug-resistant Gram-negative bacteria. They are cationic peptides that target the negatively charged lipid A component of lipopolysaccharides, disrupting the outer membrane and lysing the cell. Polymyxin resistance is conferred by inner-membrane enzymes, such as phosphoethanolamine transferases, which add positively charged phosphoethanolamine to lipid A. Here, we present the structure of MCR-1, a plasmid-encoded phosphoethanolamine transferase, in its liganded form. The phosphatidylethanolamine donor substrate is bound near the active site in the periplasmic domain, and lipid A is bound over 20 Å away, within the transmembrane region. Integrating structural, biochemical, and drug-resistance data with computational analyses, we propose a two-state model in which the periplasmic domain rotates to bring the active site to lipid A, near the preferential phosphate modification site for MCR-1. This enzymatic mechanism may be generally applicable to other phosphoform transferases with large, globular soluble domains.

The steady rise in drug-resistant Enterobacteriaceae, including *Escherichia coli* and *Klebsiella pneumoniae*, that exhibit reduced susceptibility to carbapenems and other antibiotics, constitutes a growing crisis of multidrug resistance[1,2]. To combat this epidemic more effectively, an older class of drugs known as polymyxins is employed as a last-resort treatment despite their neuro- and nephrotoxic side effects[3,4]. Polymyxins—of which polymyxin B and E (colistin) are used clinically—are cationic cyclic polypeptides. They initially target the negatively charged lipid A component of lipopolysaccharide (LPS) in the outer leaflet of the outer membrane (OM) of Gram-negative

bacteria. This and subsequent interactions with the inner membrane (IM) ultimately lyse the cells, leading to their death[5,6]. The primary mechanism used by Gram-negative bacteria to confer resistance to polymyxins involves chemical modification to LPS prior to its transport to the OM[7], affecting its net negative charge (Fig. 1a).

Resistance to polymyxins is conferred by LPS-modifying enzymes that reside in the bacterial IM. These include the 4-amino-4-deoxy-L-arabinose (L-Ara4N) transferase (ArnT), a glycosyltransferase that modifies the 1- and 4'-phosphate groups of lipid A with positively-charged L-Ara4N[8,9], and members of the phosphoethanolamine (PEtN)

[1]Department of Physiology and Cellular Biophysics, Columbia University Irving Medical Center, New York, NY, USA. [2]School of Life Sciences and Department of Chemistry, University of Warwick, Coventry, UK. [3]Department of Infectious Diseases, College of Veterinary Medicine, University of Georgia, Athens, GA, USA. [4]Department of Biochemistry and Molecular Biology, University of Chicago, Chicago, IL, USA. [5]Department of Chemistry and Biochemistry, University of Arizona, Tucson, AZ, USA. [6]Institute for Molecular Bioscience, University of Queensland, Brisbane, QLD, Australia. [7]Department of Radiation Oncology, Weill Cornell Medical College, New York, NY, USA. [8]These authors contributed equally: Allen P. Zinkle, Mariana Bunoro Batista. ✉e-mail: strent@uga.edu; rin7007@med.cornell.edu; Phillip.Stansfeld@warwick.ac.uk; fm123@cumc.columbia.edu

transferase family[10]. PEtN transferases act by adding cationic PEtN from the headgroup of the major glycerophospholipid phosphatidylethanolamine (PE) to either or both the 1- or 4'-phosphate groups of lipid A, leaving diacylglycerol (DAG) as a byproduct[11,12] (Supplementary Fig. 1a). For both ArnT and the PEtN transferases, addition of positively charged groups to lipid A reduces the net negative charge on the OM, effectively impeding polymyxins from engaging electrostatically with LPS[13,14] (Fig. 1a).

The PEtN transferases include EptA (also known as PmrC)[15,16] and MCR-1, the enzyme encoded by the first identified mobilized colistin resistance (mcr-1) gene[17]. Currently, there are ten reported mcr gene variants (mcr-1–10), among which mcr-1 is the predominant one found in clinical isolates[18]. In contrast to chromosomally encoded EptA, most MCR variants are plasmid-encoded, significantly increasing the likelihood of global dissemination through horizontal gene transfer[19,20]. Since their discovery in food, animal, and patient isolates from China in 2015[17], mcr-mediated colistin-resistant E. coli strains have been reported in over 50 countries spanning five continents[21].

PEtN transferases are structurally defined by an N-terminal transmembrane (TM) domain and a C-terminal periplasmic domain (PD). The PD is structurally defined by the presence of several central β-strands "sandwiched" between two layers of α-helices in a classical β-α-β-α motif, and contains an absolutely conserved catalytic threonine (T285 in MCR-1)[16,22,23]. Additional density present near the side chain of the active site threonine highlights two distinct and physiologically relevant states observed within the PD[23]. PEtN transferases, like EptA and MCR-1, have been proposed to utilize a $Zn^{2+}$-dependent, two-step reaction mechanism. In this process, the first step is rate-limiting and involves PEtN cleavage from PE and attachment to the catalytic threonine in the active site, forming a PEtN-Thr intermediate[12]. This is followed by a transfer of PEtN from the threonine to either the 1- or 4'-phosphate groups of lipid A, or both[11,12]. Over the last decade, several crystal structures of C-terminal domains have been determined[22–32]. A single full-length crystal structure of EptA from Neisseria meningitidis (NmEptA) has also been determined, showing that the C-terminal domain is connected by a long bridging helix (BH) to a TM domain consisting of five TM helices (TM helices 1-5)[16]. This structure provides detailed insights into the tertiary fold of PEtN transferases, and shows a dodecyl-β-D-maltoside (DDM) detergent molecule and $Zn^{2+}$ bound in a pocket spanning the periplasmic and membrane domains. However, the absence of bound native substrates limits our understanding of how this enzyme works.

Mechanistic differences are suggested to exist between EptA and MCR-1, as evidenced by the fact that chimeras—where the TM domain of one is swapped with that of the other—are functionally inactive and colistin-sensitive. In contrast, similar chimeras of MCR-1 and MCR-2 are functionally interchangeable[33]. Adding to this, while EptA from E. coli selectively modifies the 1-phosphate of lipid A, MCR-1 shows preferential selectivity towards the 4'-phosphate[34]. These findings highlight the need to obtain substrate-bound, full-length PEtN transferase structures to gain deeper insights into the mechanism underlying EptA and MCR-1–mediated modification of lipid A. A full understanding of this molecular mechanism may, in turn, provide crucial details relating to the mechanisms of other PEtN transferases that may or may not target lipid A for modification, such as the EptA homolog from Campylobacter jejuni (termed EptC or CjEptC), which can add PEtN to both LPS and the flagellar rod protein FlgG[35,36], and BcsG, which modifies cellulose with PEtN[37].

Using single-particle cryo-electron microscopy (cryo-EM) coupled with antigen-binding fragment (Fab) technology, we determined the structure of full-length MCR-1 bound to both PE and lipid A, revealing two distinct, distant ligand-binding sites. Our structural data, integrated by computational analyses and genetic, biochemical, and drug-resistance experiments, allow us to propose a unique mechanism of action for MCR-1 that involves a significant conformational change to

facilitate PEtN transfer. Furthermore, computational predictions highlight how our findings may be generally applicable to other phosphoform transferases, essential for but not limited to bacterial cell envelope synthesis, that utilize PEtN or similar glycerophospholipids as donor substrates.

## Results

### Production, functional characterization, and structure determination of MCR-1

The mcr-1 gene was incorporated into an expression vector (pMCSG7) carrying an N-terminal deca-histidine tag for metal affinity chromatography purification. MCR-1 (63.9 kDa) was subsequently expressed in E. coli, solubilized using the detergent DDM, and purified to homogeneity using metal affinity chromatography (see Methods for more details).

To assess whether the expressed protein was functional, the mcr-1 plasmid was transformed into an E. coli K-12 strain (W3110) lacking both the chromosomally encoded EptA and the gene for a second lipid A modification enzyme, LpxT, which adds an additional phosphate to the 1-phosphate of lipid A. The transformed cells were radiolabeled in media containing $^{32}P_i$, and the lipid A domain of LPS was chemically isolated, as described previously[38]. Lipid A species were analyzed by thin-layer chromatography (TLC) and visualized by phosphorimaging analysis (Supplementary Fig. 1b). This assay demonstrated that cells expressing MCR-1 produce a lipid A with a slower migration pattern compared to those carrying the vector alone, indicative of MCR-1 being able to modify lipid A by the addition of PEtN (Supplementary Fig. 1b). Comparison of the TLC pattern with previously reported data[35,39,40] confirmed that MCR-1 catalyzes only a single PEtN modification. This is in contrast to chromosomally encoded EptA from E. coli, for example, which adds two PEtN molecules, one to each lipid A phosphate group[40].

We also showed that modified lipid A could be extracted directly from purified protein and analyzed it by mass spectrometry (Supplementary Fig. 1c). Compared to lipid A extracted from whole cells lacking an endogenous PEtN transferase, which showed no modification, the lipid A species obtained from purified MCR-1 was predominantly PEtN-modified (Supplementary Fig. 1c).

To mimic a native-like bilayer environment, detergent-purified MCR-1 was reconstituted into lipid-filled nanodiscs. From a reconstitution screen comparing the elution profiles of purified MCR-1 incorporated into each of six distinct nanodiscs to that of MCR-1 in DDM, a nanodisc composed of 1-palmitoyl-2-oleoyl-sn-glycero-3-phospho-(1'-rac-glycerol) (POPG) lipid was chosen, due in part to the homogeneity of the elution profile and its shift leftward relative to MCR-1 in DDM (see Supplementary Information for further details). Reconstitution into POPG-filled nanodisc (Supplementary Fig. 1d, e) was performed prior to structure determination by cryo-EM. To overcome size limitations and to provide fiducials for alignment[41], a synthetic phage display library was screened to select recombinant Fabs targeting MCR-1[42]. Among seven high-affinity Fab candidates (MR1-MR7) screened for complex formation with MCR-1 (Supplementary Fig. 2a), MR6 was identified as a top binder along with MR1 and MR5 (Supplementary Fig. 2b, c). However, MR6 was ultimately chosen based on two-dimensional (2D) class averages obtained from preliminary cryo-EM screening.

Following sample vitrification and cryo-EM analysis based on data processing from particles picked from 5465 micrographs (see Methods for further details), a density map was obtained and locally refined around the TM domain and PD of MCR-1 and MR6, achieving a global resolution of 3.6 Å (Supplementary Table 1, Supplementary Fig. 3). Using Namdinator[43], an initial AlphaFold2 model[44] of MCR-1 was flexibly fitted into the density map, with subsequent refinement resulting in a structure comprising residues 10–541 (Fig. 1b, Supplementary Fig. 4).

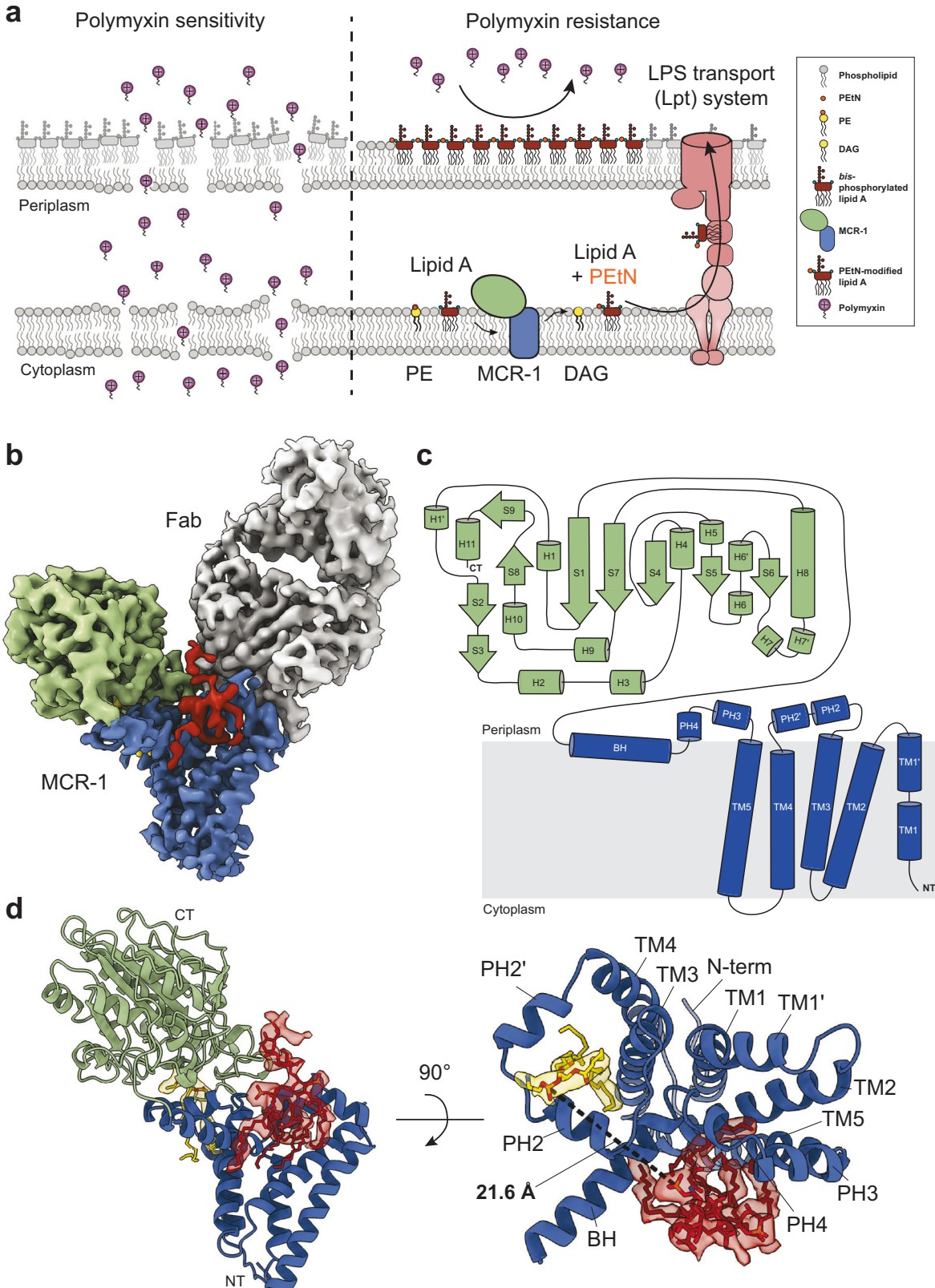

## Structure of MCR-1

The overall structure of MCR-1 closely resembles the *Nm*EptA crystal structure, with the greatest deviation occurring along the BH region (Cα RMSD value of 2.4 Å)[16] – for instance, several lysines present in the BH of MCR-1 (K204, K211, K212) are replaced with non-charged residues in EptA (Supplementary Fig. 5a, b). The structure includes an N-terminal domain of five TM helices (TM helices 1–5) and a C-terminal soluble globular PD (Fig. 1c). The first TM helix is split into two shorter segments (TM helix 1 and TM helix 1'). Similarly, TM helices 2-4 are all relatively short in length, with only the fifth TM helix long enough to span the membrane. Linking the TM helices are two short periplasmic-facing linkers, each consisting of a pair of short α-helices – one pair

**Fig. 1 | The role of MCR-1 in conferring polymyxin resistance in Gram-negative bacteria and the structure of MCR-1. a** Schematic representation of polymyxin sensitivity, on left, and polymyxin resistance, on right, showing modification of lipid A (dark red) with PEtN (small orange circle) by MCR-1 (blue/green) and the transport of modified lipid A from the inner membrane to the outer membrane, electrostatically repelling polymyxins. PE donor substrate is shown as a yellow circle with a smaller orange circle and with two lines to represent acyl chains. Individual polymyxin molecules are depicted as purple spheres attached to a small tail and carrying a positive charge. **b** Cryo-EM density map of the PE- and KLA-bound MCR-1–Fab complex. Density corresponding to the Fab is shown in gray, and the PD and TM domain of MCR-1 are colored green and blue, respectively. The density for KLA is in red, and the density for PE in yellow. **c** Schematic diagram showing the topology of MCR-1, which consists of five TM helices and a BH connecting the N-terminal TM domain to a large C-terminal PD. Two periplasmic-facing loops, each containing two small α-helices, are present between TM3/4 (PH2/PH2') and TM5/BH (PH3/PH4). **d** The 3.6 Å cryo-EM structure of PE- and KLA-bound MCR-1 shown in two different orientations, colored as in (**c**) The N and C termini are labeled. On the right, the structure has been rotated 90°, as viewed from the top, with helices numbered and the PD removed. PE is shown as sticks in yellow under semi-transparent density. KLA, colored red, is also shown as sticks and under semi-transparent density. The distance measured between the phosphate of PE and the nearest phosphate group belonging to KLA (1-phosphate) is shown as a black dashed line. Nitrogen, oxygen, and phosphorus atoms shown in sticks are colored blue, red, and orange, respectively.

(periplasmic helix (PH) 2 and PH2') located between TM helices 3 and 4, and the other (PH3 and PH4) between TM helix 5 and the BH connecting the TM and PD. $Zn^{2+}$ ions shown in several di-zinc crystal structures of the MCR-1 PD[27,28] and the mono-zinc *Nm*EptA crystal structure[16], do not appear to be present. The MR6 Fab is bound on the periplasmic side of the membrane adjacent to the PD above the TM domain, behind PH3 and PH4 and above TM helices 1' and 2 (Fig. 1b). Comparison of the MCR-1 and *Nm*EptA structures shows no significant conformational changes are induced by Fab binding (Fig. 1b, Supplementary Fig. 5a).

## Putative substrate binding sites

Two additional densities were observed, which, based on their shape, size, and location were unambiguously identified as PE and $Kdo_2$-lipid A (KLA) and built into the map (Fig. 1d). The PE density appears as a small hairpin-like shape located near the conserved catalytic nucleophile (T285), within the predicted active site pocket of MCR-1 formed between PH2 and PH2' (Fig. 1d). The upper portion of the PE density exhibits a shape that accommodates the PEtN headgroup (Fig. 2a, Supplementary Fig. 4). Situated nearby are a cluster of conserved residues, including the charged E116, E246, and K333; polar H395, H466, H478; and the catalytic T285 (Fig. 2a). Consistent with our characterization of this density, liquid chromatography-mass spectrometry (LC-MS) analysis of MCR-1 reconstituted in POPG-filled nanodiscs showed the presence of multiple PE species (Supplementary Fig. 6).

To further study the PE binding site, ten conserved residues near the identified density (M105, N108, T112, E116, E246, T285, K333, H395, H466, and H478) were individually mutated to alanine. Protein expression in *E. coli* was verified (Supplementary Fig. 7a), and the activity of each MCR-1 variant was assessed using TLC to analyze the [$^{32}$P]-lipid A species synthesized by the cells. The assay revealed that six mutations (E116A, E246A, T285A, H395A, H466A, and H478A) completely abolished activity, while two (M105A and K333A) significantly reduced it (Fig. 2b). The remaining two mutations (N108A and T112A) also diminished activity, albeit to a lesser extent (Fig. 2b, Supplementary Fig. 7b). These results align with previous biochemical studies investigating amino acid substitutions and their influence on sensitivity of MCR-1 to colistin and polymyxin B[29,33,45–47]. They also corroborate structural data from studies on EptA and MCR-1, highlighting the critical roles of residues such as E246, H395, H466, H478, and the catalytic nucleophile T285 for activity[16,22–24,27–31].

A larger, multi-armed density lies over 20 Å from the PE density, situated directly adjacent to the BH, atop TM helices 2 and 5, and in front of PH3 and PH4 (Fig. 1d). The shape of the density accommodates a KLA molecule, including both the 1- and 4'-phosphate groups, the two Kdo sugars, and four of the six acyl chains (Fig. 2a, Supplementary Fig. 4), and was modeled accordingly. KLA is surrounded by several basic or charged (or both) residues with varying levels of conservation (Supplementary Fig. 5b). These include D102 on PH2, R184 on PH3, K187 and R190 on PH4, and K204 on the BH (Fig. 2a). R184 is adjacent to the 4'-phosphate of KLA, while the other three residues are closer to the 1-phosphate (Fig. 2a). This came as a surprise, as KLA had been previously suggested to bind closer to PE[16,48,49].

Our mass spectrometry analysis of lipid A isolated from purified MCR-1 revealed the presence of both PEtN-modified and unmodified lipid A (Supplementary Fig. 1c). This observation suggests a mixture of modified and unmodified particles may exist within the cryo-EM dataset. However, we modeled unmodified KLA due to the absence of additional density near either the 1- or 4'-phosphates.

To investigate the KLA binding site, we performed alanine site-directed mutagenesis of the five basic residues near its upper portion (D102, R184, K187, R190, and K204), as well as two hydrophobic residues near the modeled acyl chains (F34 (TM helix 1') and L58 (TM helix 2)) (Supplementary Fig. 7a). After confirming protein expression, the lipid A species were isolated and analyzed from cells making each of the mutants (Fig. 2b).

Compared to wild-type MCR-1, which led to PEtN modification of 85% of the isolated lipid A, R190A showed a significant reduction in activity, with only 28% of the lipid A species being PEtN-modified (Fig. 2b, Supplementary Fig. 7b). Two other mutations, D102A and R184A, also decreased activity, though to a lesser extent, with 66 and 67% of lipid A being modified, respectively (Fig. 2b, Supplementary Fig. 7b).

To explore whether specific residues compensate for the loss of others, we generated four double-mutants, each incorporating the R190A mutation along with either D102A, R184A, K187A, or K204A. All double mutants exhibited a greater reduction in MCR-1 activity compared to any of the single ones, including R190A alone. Notably, the K187A/R190A double mutant almost completely abolished activity (Fig. 2b, Supplementary Fig. 7b).

## Polymyxin susceptibility from select mutants

To investigate the effect of select MCR-1 mutations on polymyxin resistance in vivo, efficiency-of-plating assays were conducted, and minimum inhibitory concentrations (MICs) of polymyxin B (PMB) were measured (Fig. 2c, Supplementary Fig. 7c). Select mutations targeting residues in the binding sites for both PE (M105A, T112A, E246A, K333A, and H395A) and KLA (K187A, R190A, and K187A/R190A) were assessed (Fig. 2c).

From the efficiency-of-plating assays, the expression of each of the mutants tested resulted in PMB sensitivity, except for K187A, which, similarly to the WT, displayed PMB resistance even at a PMB concentration of 1.5 µg/mL (Fig. 2c). This is not unexpected, given that the K187A mutation alone does not appear to affect MCR-1 activity to a significant degree (Fig. 2b). The MICs for WT MCR-1 (3.0 µg/mL) and the K187A mutant (2.0 µg/mL) are also comparable in their display of PMB resistance (Supplementary Fig. 7c). Cells expressing KLA binding site mutations previously shown to significantly reduce or abolish activity (R190A, K187A/R190A) (Fig. 2b) were sensitive to PMB (Fig. 2c, Supplementary Fig. 7c).

Cells expressing mutants targeting the PE binding site were generally sensitive to PMB (Fig. 2c, Supplementary Fig. 7c), although T112A, which was shown to only moderately reduce MCR-1 activity

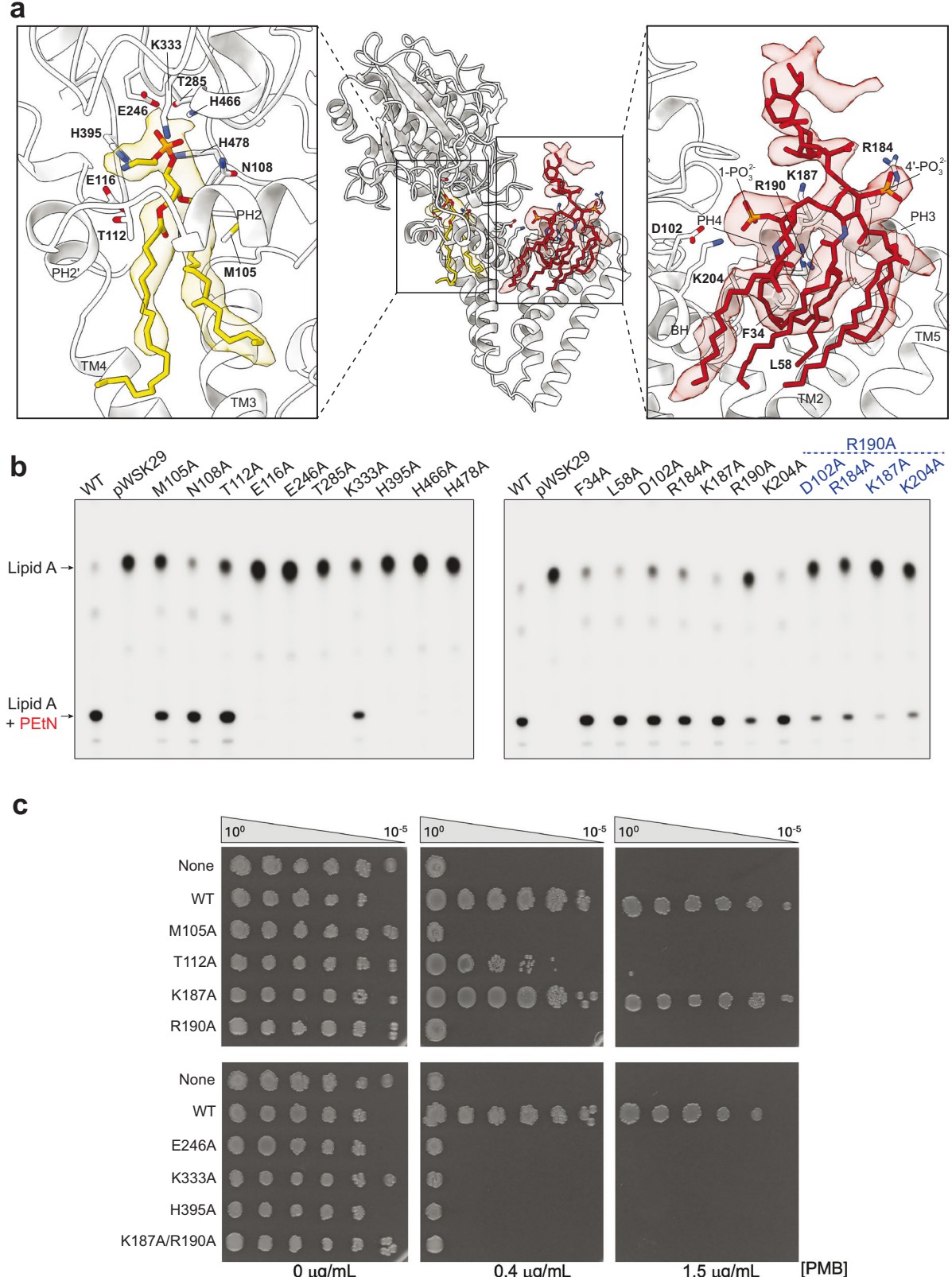

**Fig. 2 | Binding sites for PE and lipid A with functional validation. a** On the left, a zoomed in panel depicting the PE binding site within the MCR-1 structure, in cartoon representation and colored light gray. PE is shown and colored as in Fig. 1d, and side chains of conserved residues in close proximity to the ligand are shown as sticks and colored in accordance with the structure. On the right, a zoomed in panel depicting the KLA binding site between the BH and PH3/4. KLA is shown and colored as in Fig. 1d, and side chains of nearby residues are shown as sticks. Nitrogen, oxygen, and phosphorus atoms are colored as in Fig. 1d. **b** TLC plate showing the migration of lipid A isolated from W3110 (Δ*lpxT*, Δ*eptA*) cells grown in LB broth and transformed with MCR-1 WT, empty pWSK29 plasmid, and the indicated MCR-1 mutants. Double mutants with R190A are denoted by blue labeling. Protein expression was induced with 10 μM IPTG. Major $^{32}$P-labeled lipid A species are indicated with an arrow and labeled to the left. The data represent three biological replicates. **c** Efficiency-of-plating assay depicting 10-fold dilutions of cells expressing MCR-1 WT, mutants, or neither, labeled to the left, in the presence of 0, 0.4, and 1.5 μg/mL PMB.

(Fig. 2b), maintains some degree of resistance to PMB (Fig. 2c). However, sensitivity can begin to be observed at a PMB concentration of 0.4 µg/mL, with cell growth completely inhibited at 1.5 µg/mL (Fig. 2c). When looking at PMB MICs alone, T112A is indistinguishable from the control cells lacking MCR-1 and the other mutants displaying PMB sensitivity, with each having a PMB MIC less than 0.064 µg/mL (Supplementary Fig. 7c). These results provide in vivo evidence that residues in the two identified substrate binding sites are either important or essential for the physiological activity of MCR-1.

### Conformational flexibility of the MCR-1 PD

The proposed two-step reaction mechanism for MCR-1 involves first the formation of a PEtN-T285 intermediate and second the transfer of PEtN to either phosphate group of KLA[12,50]. Based on the relative positions of PE and KLA in our structure, this mechanism would require a conformational change in the PD of MCR-1 to bridge either the ~22 Å (1-phosphate) or ~33 Å (4′-phosphate) gap between the KLA phosphates and T285 (Fig. 1d). To evaluate whether the PD might adopt a conformation different from that observed in our structure, we used AlphaFold2[44] to generate five computational models of MCR-1. While three of the models adopt a conformation similar to our structure and the *Nm*EptA crystal structure (hereafter described as State 1), the other two predictions display an alternative conformation (State 2). In State 2, the PD is rotated along the BH and positioned adjacent to the KLA binding site, between PH3 and PH4 (Fig. 3a, Supplementary Fig. 8). This alternate conformation aligns T285 proximal to the 4′-phosphate of KLA. In this state, the oxygen of the T285 side chain is positioned just 7.2 Å away from the nearest oxygen of the 4′-phosphate (Fig. 3b).

To explore potential conformational transitions of MCR-1, we conducted atomistic molecular dynamics (MD) simulations of MCR-1 in its apo state, embedded in a bacterial membrane model composed of POPE:POPG. Simulations were initiated from both our experimental structure (State 1) and the AlphaFold2 prediction (State 2). To monitor the dynamics of the PD, we measured the change in angle over time between two intersecting planes defined by three residues taken from the BH, PD, and TM domain (Supplementary Fig. 9a). The angle between the two planes is close to -25° for State 1, around 50° for State 2, and 0° when the PD is directly above the BH. For State 1, the angle remains close to -25° across four out of five repeats, with one simulation exhibiting a transient rotation past the BH (0°) toward the State 2 conformation before returning to State 1 (Fig. 3c – orange curve, Supplementary Fig. 9b, Supplementary Movie 1). In this apo configuration, State 2 was less stable, with all five simulations displaying greater variance in angle changes over time compared to State 1 (Fig. 3c).

To study the influence of the bound ligands in the simulations, we simulated MCR-1 in State 1 in the presence of PE and KLA (Supplementary Fig. 9c). The addition of PE and KLA reduces the degree of rotation of the MCR-1 PD; however, a partial transition is still observed in one repeat from State 1. This behavior appears to be due to the instability of the PE in the binding site (Supplementary Fig. 9c). Previous structures have captured a bound $Zn^{2+}$ ion in the active site. The inclusion of $Zn^{2+}$ in the atomistic simulations stabilizes the bound PE, and in doing so reduces the conformational changes of the PD (Supplementary Fig. 9d, Supplementary Fig. 10a, b, Supplementary Movie 2).

To capture the different stages of the proposed two-step reaction mechanism, we also modeled the PEtN-T285 intermediate in the presence of KLA and $Zn^{2+}$ for both State 1 and State 2. While the PD flexibility increased compared to the ligand-bound unmodified enzyme, no transition toward State 2 was observed in simulations initiated from State 1 (Supplementary Fig. 9e). In State 2, the addition of KLA and $Zn^{2+}$ to the intermediate enzyme appears to stabilize this conformation (Supplementary Fig. 9e, Supplementary Fig. 10c, d, e, f).

Moreover, this PEtN-T285 intermediate reproducibly stabilizes at an approximate distance only 4.8 Å from the 4′-phosphate of KLA over the course of the simulations (Supplementary Fig. 11).

### Site-of-modification selectivity preference for MCR-1

Previous studies have indicated differing preferential modification sites on KLA between MCR-1 and EptA from various species. For example, EptA from *Salmonella enterica* (*Se*EptA) and *E. coli* (*Ec*EptA) preferentially modify the 1-phosphate[40], whereas EptA from *Pseudomonas aeruginosa* (*Pa*EptA) and MCR-1 selectively modify the 4′-phosphate position, respectively[34,51]. To validate the predicted preference of MCR-1 for the 4′-phosphate and to elucidate differences in preference between MCR-1 and EptA, we removed either the 1- or 4′-phosphate group of lipid A by heterologous expression of the phosphatases LpxE or LpxF, respectively[52,53]. We then assessed the ability of MCR-1 and *Ec*EptA to modify lipid A by TLC, using *Pa*EptA as a control. Consistent with previous reports[40], *Ec*EptA was capable of modifying either or both the 1- and 4′-phosphates of lipid A, without a clear preference. In contrast, MCR-1 is only able to modify the 4′-phosphate (Supplementary Fig. 12a, b). This agrees with both the KLA's 4′-phosphate position in our structure as well as the AlphaFold2 prediction for State 2 (Fig. 3a, b) and our computational analysis showing the PEtN-Thr285 is stabilized with the PEtN located less than 5 Å from the KLA 4′-phosphate (Supplementary Fig. 11).

### Salt-bridge disruption promotes MCR-1 conformational change

To investigate the interactions between the TM domain and PD, we performed a categorical Jacobian calculation to extract the coevolutionary signal from the MCR-1 sequence. This analysis revealed abundant intra-domain contacts within both the TM domain and PD but identified only a single predicted inter-domain contact, namely, a salt-bridge between D119 and R402 (Supplementary Fig. 13a).

Guided by this finding, we analyzed salt-bridge interactions between the TM domain and PD (Fig. 3d). Our analyses consistently revealed a salt-bridge between the previously identified D119 and K401 (Fig. 3b), the residue adjacent to R402. This salt-bridge was disrupted during the apo simulations that partially transitioned from State 1 to State 2 (Fig. 3c, d), suggesting that this interaction plays an important role in stabilizing State 1.

To test this hypothesis, we performed in silico mutagenesis to generate alanine mutants of D119, K401, and R402, respectively. Simulations of the K401A mutant consistently demonstrated a complete transition from State 1 to State 2 (Fig. 3e, Supplementary Movie 3). Similarly, the R402A mutant also promoted this transition, though at a lower frequency (Supplementary Fig. 13b). In contrast, no transitions were observed for D119A (Supplementary Fig. 13c). This finding underscores the role of the salt-bridge between the TM domain and PD as a likely important, if not critical, molecular lock. Unlocking this interaction is necessary to enable the conformational transition from State 1 to State 2, thereby facilitating PEtN transfer.

Taken together, our results validate the two-step reaction mechanism proposed for PEtN transferases like MCR-1[12,28,33,50] (Fig. 4). Specifically, the first step consists of PE donor substrate binding, triggering PEtN cleavage to produce DAG. This disrupts the conserved salt-bridge tethering the PD in the State 1 conformation, while producing an enzyme intermediate with PEtN bound to the catalytic nucleophile T285. In the second step, following KLA acceptor substrate binding, the PD of PEtN-T285 becomes stabilized in close proximity to the 4′-phosphate of KLA, enabling selective modification at that site (Fig. 4).

### Similarities with other phosphoform transferases

This proposed mechanism for MCR-1 is likely to apply to other phosphoform transferases predicted to have major soluble domains,

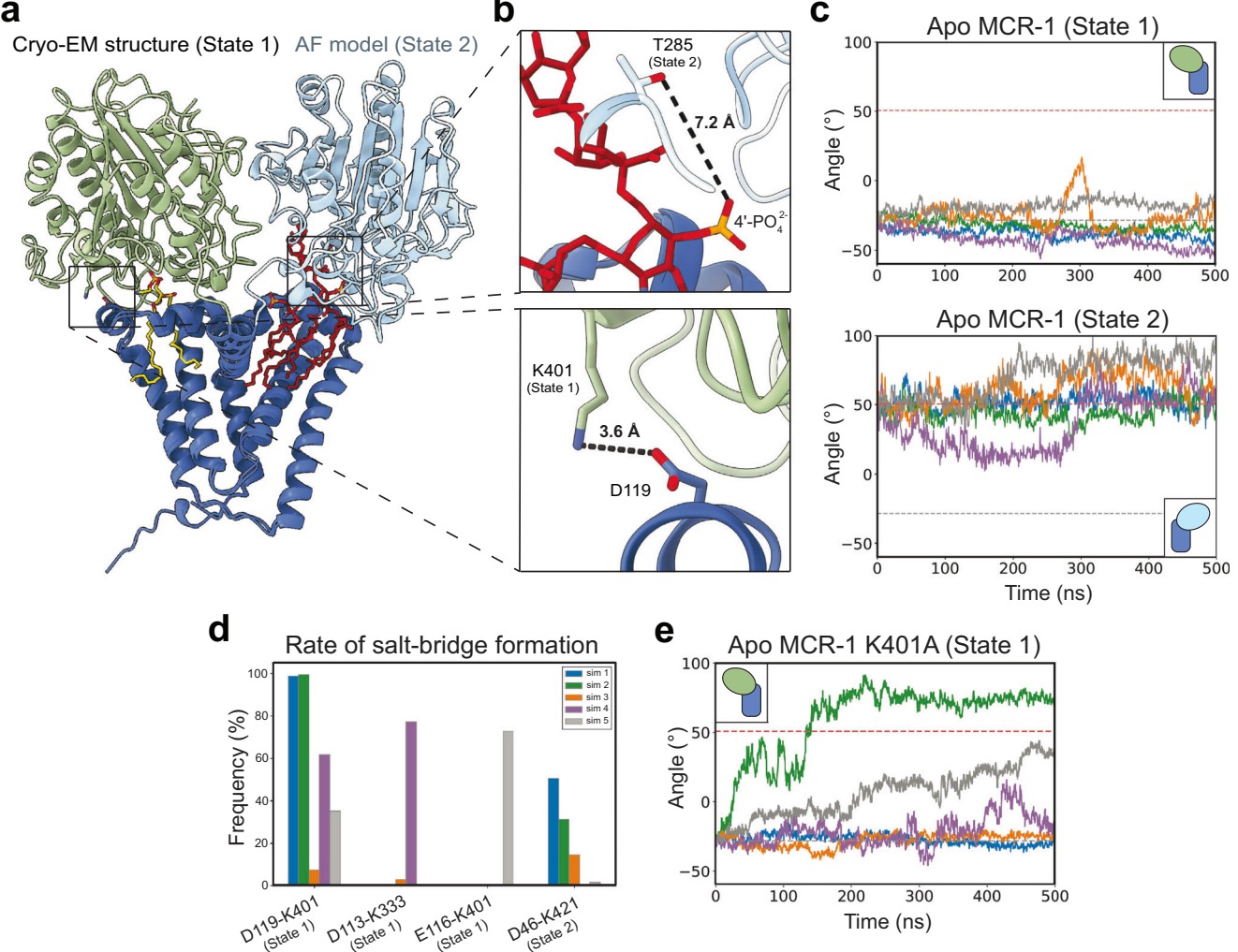

**Fig. 3 | Flexibility of the MCR-1 periplasmic domain. a** The MCR-1 cryo-EM structure, colored as in Fig. 1, is overlaid along the TM domain with an AlphaFold2 (AF) model of MCR-1 in the State 2 conformation. The TM domain of the AF model is colored the same as that of the cryo-EM structure, while the PD is colored cyan. **b** On top, a zoomed in panel showing the distance between the catalytic nucleophile T285 in the State 2 model and the 4′-phosphate group of KLA in the cryo-EM structure. T285 and KLA are shown as sticks, with KLA colored as in Fig. 1. On the bottom, a zoomed in panel of the salt-bridge, predicted to form in the State 1 conformation, between K401 in the PD and D119 in the TM domain. Side chains for K401 and D119 are shown as sticks and colored in accordance with the cryo-EM structure in (**a**). Nitrogen, oxygen, and phosphorus atoms are colored blue, red, and yellow, respectively. Distances between side chains are shown as dotted black lines and labeled. **c** On top, five repeats of atomistic MD simulations over 500 ns performed with apo MCR-1 in the State 1 conformation, depicted in the top right corner as a cartoon colored as in Fig. 1a. On the bottom, five repeats of atomistic MD

simulations over 500 ns performed with apo MCR-1 in the State 2 conformation, depicted in the bottom right corner as a cartoon with the PD and TM domain colored in accordance with the AF model in (**a**). The simulations are colored gray, green, orange, purple, and blue, respectively. The dashed gray and red lines correspond to the angle for the State 1 and State 2 conformations, respectively. **d** Frequency of the salt-bridges forming between the PD and the TM domain for the apo State 1 and State 2 conformations. Salt-bridges that formed at a frequency higher than 50%, in at least one of the simulation repeats, are included in the analysis. **e** Five repeats of atomistic MD simulations over 500 ns performed with apo K401A MCR-1 mutant in the State 1 conformation, depicted in the top left corner as a cartoon colored as in Fig. 1a. A full transition from the State 1 conformation to the State 2 conformation is observed for 2 repeats (green and gray curves), and a partial transition is observed for one repeat (purple curve). Source data are provided as a Source Data file.

including EptB, *E. coli* EptC, and BcsG[37] (Supplementary Fig. 14), among others. Like MCR-1 and EptA, these and other phosphoform transferases also bind large acceptor substrates. In the case of EptB, for example, PEtN is transferred to the Kdo sugars of KLA[54], whereas in the case of *E. coli* EptC, modification occurs on a heptose core sugar of the larger lipid A-core oligosaccharide complex[55]. Other large acceptor substrates modified by phosphoform transferases include periplasmic glycans like cellulose, which are modified with PEtN by BcsG, as well as OPGs (modified with phosphoglycerol by OpgB)[56] and LTA (formed by phosphoglycerol addition by LtaS to the growing poly-phosphoglycerol chain)[57]. Two-state AlphaFold3 models of EptB,

EptC, and BcsG from *E. coli* show how the soluble domains for each of these enzymes can rotate to enable selective PEtN modification at distinct sites on their respective substrates – with EptB/C, on the Kdo or core sugars of lipid A, and with BcsG, on the larger cellulose (Supplementary Fig. 14). From the MCR-1 structure, multiple charged residues (D102, R184, K187, R190, and K204) can be seen surrounding the phosphate groups of the lipid A acceptor and are critical for function (Fig. 2). The acceptor binding sites and bridging helices between the soluble and TM domains of other phosphoform transferases may similarly harbor charged residues to help coordinate specific functional groups unique to their cognate substrates.

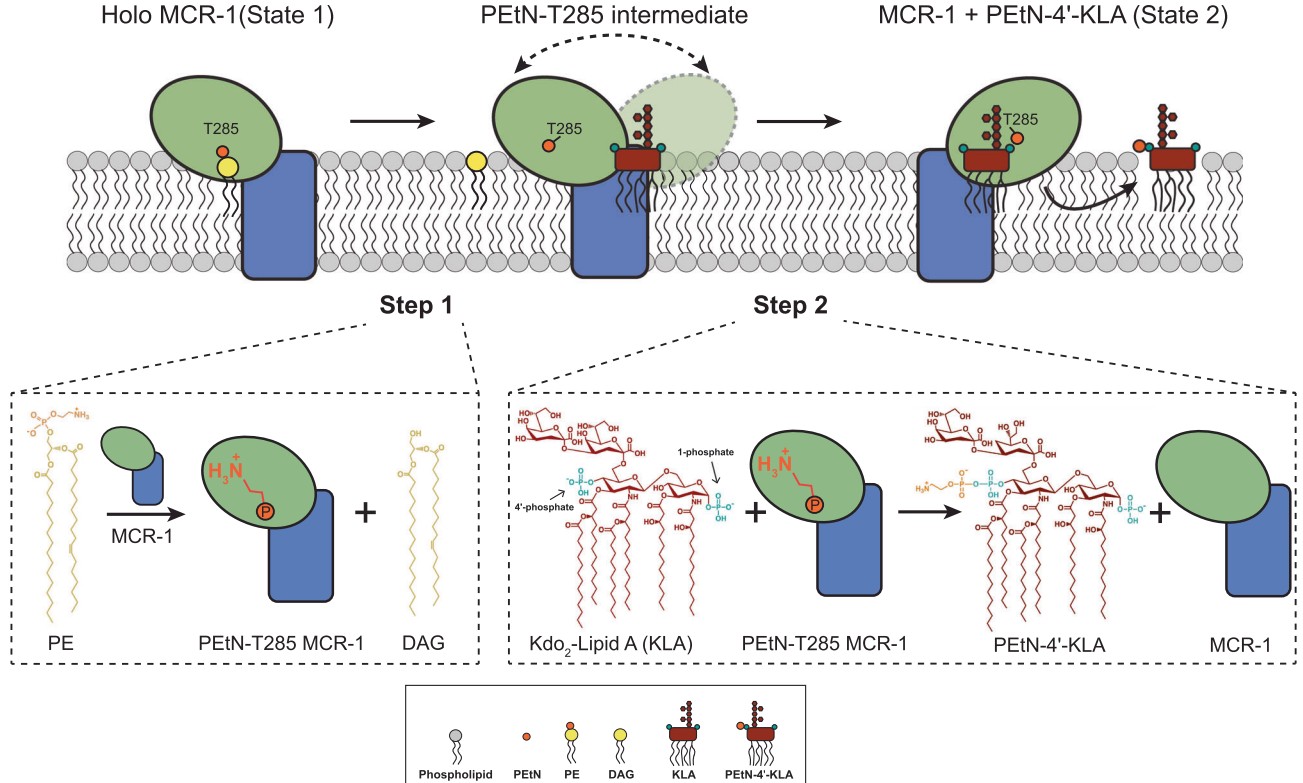

**Fig. 4 | Two-step mechanism of action of MCR-1.** Schematic representation of the two-step reaction mechanism MCR-1 undergoes to modify the 4'-phosphate of KLA. From left to right, MCR-1, shown and colored as in Fig. 1, in the State 1 conformation, cleaves PEtN from PE, which is depicted as in Fig. 1a. The PEtN-bound T285 intermediate enzyme is then able to rotate the PD to position itself behind KLA, also depicted as in Fig. 1a. In State 2, PEtN-bound T285 is oriented to enable chemical modification of the 4'-phosphate group of KLA prior to its release. Below, steps 1 and 2 of the two-step reaction mechanism are drawn, with PE, DAG, KLA, and PEtN-4'-KLA depicted by their chemical structures and with the PEtN moiety labeled in red. MCR-1 is shown as a cartoon depicted and colored as above, and the PEtN-T285 intermediate is shown with the PEtN moiety drawn as an orange line connected to a small orange circle in the PD.

## Discussion

Rising multidrug resistance has entailed the greater therapeutic use of last-resort antibiotics, such as polymyxins. PEtN transferases confer resistance to polymyxins by removing cationic PEtN from a donor PE in the IM and transferring it to one or both phosphates of lipid A. PEtN-modified lipid A reduces the overall net negative charge of the OM, compromising polymyxin engagement. Understanding PEtN transferase-mediated mechanisms of polymyxin resistance is, therefore, of paramount importance.

We have determined the full-length structure of a plasmid-born PEtN transferase – namely, that of substrate-bound MCR-1 – revealing the lipid A binding site. The cryo-EM structure shows that PE and lipid A bind at two distinct sites, separated by over 20 Å. This finding, which we validate with extensive functional and polymyxin-resistance assays on select mutants, comes unexpected[16,48,58] and leaves open the intriguing question of how this enzyme works.

Using AlphaFold2 predictions and MD simulations, we show that MCR-1 likely undergoes a major conformational change involving rotation of the PD from the PE-binding site to the opposite side of the BH, where KLA is located. In support of this, we demonstrated biochemically that MCR-1 preferentially modifies the 4'-phosphate KLA, the furthest phosphate from the PE in our structure. Furthermore, our atomistic MD simulations provide a mode for this conformational switch. We propose that the BH acts as a vaulting table to enable the movement of the PD between states. This conformational change enables the long-distance transfer of PEtN from the modified T285 to the 4'-phosphate of lipid A. Predicted structures of other PEtN transferases, including BcsG, in two distinct conformations (Supplementary Fig. 14) underscore the likelihood that a similar mechanism may be utilized by other phosphoform transferases with globular soluble domains to modify their respective substrates.

The reaction catalyzed by MCR-1 is thought to be $Zn^{2+}$-dependent. While we do not observe bound $Zn^{2+}$, we cannot rule out the possibility that $Zn^{2+}$ is present – indeed, for the first of the two previously identified $Zn^{2+}$ binding sites[28], limited local resolution prevents distinguishing between side chain densities for E246 and T285 and any possible $Zn^{2+}$ density in the cryo-EM map; and the second $Zn^{2+}$ binding site overlaps with PE in our structure. Given that this second $Zn^{2+}$ is predicted to be required only for the transfer of PEtN to lipid A (and not cleavage of PEtN from PE)[12], it is possible this $Zn^{2+}$ is not recruited until after PEtN has been cleaved from PE. Regardless, MD simulations show that the inclusion of $Zn^{2+}$ reduces the flexibility of the PD in both State 1 and 2 conformations and stabilizes ligand binding (Supplementary Fig. 9b, d, Supplementary Movie 2). The MR6 Fab, required for structure determination, binds directly above the TM domain and adjacent to the PD in our structure. This likely hinders, if not impedes, the movement of the PD from State 1 to State 2, thereby explaining a lack of State 2 particles in the cryo-EM dataset. It could also explain the fact that KLA appears unmodified in our structure. MR6 binds on the periplasmic side of the membrane, between PH3 and a small periplasmic-facing loop between TM1' and TM2. Comparison between the MCR-1 cryo-EM structure and the *Nm*EptA crystal structure shows no apparent small or major conformational changes induced by MR6 binding. Minor alterations between the EptA and MCR-1 structures can be seen within the small loop bridging TM1' and TM2, but this is likely attributable to the presence of an additional three residues within the loop in EptA compared to MCR-1.

Our results also highlight differences in lipid A modification between MCR-1 and EptA. While MCR-1 and *Pa*EptA selectively modify the 4′-phosphate, other EptA homologs including *Ec*EptA, *Nm*EptA, *Se*EptA, *Cj*EptC, and *Vibrio cholerae* (*Vc*EptA) can add PEtN to both phosphate groups of lipid A[15,35,39,40]. Sequence comparison between MCR-1 and *Pa*EptA (which selectively modify only the 4′-phosphate) to those belonging to the other EptA orthologs (which modify both phosphates) reveals key differences. Among the KLA-binding residues critical for function (D102, R184, K187, R190, and K204) (Fig. 2b), K204 is only conserved in MCR-1 and *Pa*EptA (Supplementary Fig. 5b). In our structure, the side chain of K204, which is located on the BH, reaches in close proximity to the 1-phosphate of KLA (Fig. 2a). The absence of K204 in EptA orthologs capable of modifying both phosphates may therefore permit KLA to bind in one of two different orientations – in one case, as seen here, with the 1-phosphate closest to the BH; and in the other case, rotated 180° such that the 4′-phosphate is adjacent to the BH. Future structural work, particularly with respect to KLA-bound EptA orthologs, is required to shed light on these differences.

Our work provides a framework for understanding how PEtN and other phosphoform transferases function and could offer a structural basis for designing inhibitors of this mechanism for conferring polymyxin resistance.

## Methods

### Target identification and cloning

Ligation-independent cloning (LIC) was used to clone *mcr-1* from *E. coli* into an LIC-adapted expression vector (pMCSG7[59]) bearing an N-terminal protease-cleavable decahistidine tag for metal-affinity-chromatography-based purification (Supplementary Table 2). Small-scale expression and purification tests, including detergent screens, were performed as previously described[60]. All cloning and initial protein characterization experiments were performed at the former Center on Membrane Protein Production and Analysis (COMPPÅ) at the New York Structural Biology Center (NYSBC).

### Protein expression and purification in detergent

MCR-1, cloned in the pMCSG7 vector, was used to transform BL21(DE3) pLysS *E. coli* competent cells and grown overnight in 2xYT medium supplemented with $100 \, \mu g \, mL^{-1}$ ampicillin and $35 \, \mu g \, mL^{-1}$ chloramphenicol at 37 °C with shaking (200 revolutions per minute (rpm)). The next day, 800 mL (for large-scale protein expression) or 31.5 mL (small-scale to test expression) of the same medium was inoculated with the starter culture at a 1:100 ratio and left to grow at 37 °C with shaking (200 rpm) until the optical density at 600 nm ($OD_{600 \, nm}$) reached 0.6–0.8. The temperature was then reduced to 22 °C, protein expression was induced with 0.2 mM isopropyl β-D-thiogalactopyranoside (IPTG), and the culture was incubated overnight with shaking (200 rpm). Cells were collected by centrifugation (6371 x g for 25 min) at 4 °C, washed once with phosphate-buffered saline (PBS), and centrifuged again (1959 x g for 30 min) to produce a pellet that was stored at -80 °C until further use. For large-scale purification of MCR-1, cell pellets were resuspended in lysis buffer containing 20 mM HEPES pH 7.5, 200 mM NaCl, 20 mM MgSO₄, $10 \, \mu g \, mL^{-1}$ DNase I, $8 \, \mu g \, mL^{-1}$ RNase A, 1 mM TCEP, 1 mM PMSF, and Complete Mini EDTA-free protease inhibitor cocktail (Roche) used according to the manufacturer's instructions. Cells were lysed with an Emulsiflex C3 homogenizer (Avestin), and the lysate was solubilized for two hours with *n*-dodecyl-β-D-maltopyranoside (DDM; Anatrace) added to a final concentration of 1% (w/v) in a volume of approximately 40 mL per cell pellet from 800 mL culture. Insoluble material was removed by ultracentrifugation at 134,392 x g for 30 min at 4 °C and the protein was purified from the supernatant by metal-affinity chromatography. The supernatant, after addition of imidazole to a final concentration of 40 mM, was incubated with pre-equilibrated Ni-NTA agarose beads (Qiagen) (0.7 mL per pellet from an 800 mL culture) overnight. The

beads were then loaded on a column and washed with ten column volumes of 20 mM HEPES pH 7.5, 500 mM NaCl, 75 mM imidazole, and 0.03% (w/v) DDM. Protein was eluted with four column volumes of 20 mM HEPES pH 7.5, 150 mM NaCl, 300 mM imidazole, and 0.03% (w/v) DDM. Imidazole was removed from the pooled elution fractions by exchanging buffer to 20 mM HEPES pH 7.5, 150 mM NaCl, and 0.03% (w/v) DDM (final protein buffer) using a PD-10 desalting column (GE healthcare).

### Nanodisc incorporation after detergent purification

MCR-1 was incorporated into lipid nanodiscs with a 1:200:5 molar ratio of protein:POPG:membrane scaffold protein 1D1 (MSP1D1)[61,62]. This mixture was incubated at 4 °C for two hours with gentle agitation. Reconstitution was initiated by removing detergent with the addition of 100 mg Bio-beads (Bio-Rad) at 4 °C overnight with constant rotation. The nanodisc reconstitution mixture was then bound again to Ni²⁺-NTA agarose beads (0.4 mL per sample originating from 800 mL culture) at 4 °C for four hours to remove empty nanodiscs. The resin was washed with 10 column volumes of wash buffer (20 mM HEPES pH 7.5, 150 mM NaCl, and 60 mM imidazole) followed by four column volumes of elution buffer (20 mM HEPES pH 7.5, 150 mM NaCl, and 300 mM imidazole). The protein was further purified by loading it onto a Superdex 200 Increase 10/300 GL size-exclusion column (GE Healthcare Life Sciences) equilibrated in gel filtration buffer (20 mM HEPES pH 7.5 and 150 mM NaCl). Protein typically eluted as a sharp monodispersed peak, observed by monitoring the absorbance at 280 nm (Supplementary Fig. 1d).

### MCR-1 complex formation with the Fab MR6

Nanodisc-incorporated MCR-1 was incubated with the MR6 Fab at 4 °C for two hours in a 1:3 molar ratio of protein to Fab. The MCR-1–Fab complex was concentrated and filtered, then loaded on a Superdex 200 Increase 10/300 GL size-exclusion column (GE Healthcare Life Sciences) in gel filtration buffer (20 mM HEPES pH 7.5 and 150 mM NaCl).

### Phage display to identify MCR-1–specific Fab fragments (MR6)

MCR-1 was reconstituted into nanodiscs formed using chemically biotinylated MSP1D1, which were prepared as previously described[42], and the efficiency of biotinylation was evaluated by capturing the complex on streptavidin-coated paramagnetic particles (Promega). Fab selections were conducted using Fab Library E, starting with 200 nM nanodisc-reconstituted MCR-1 in the first round and decreasing to 10 nM by the fifth and final round to enhance selection stringency, as previously described[42,63–65]. The initial round was performed manually, whereas rounds 2–5 were semi-automated using a King-Fisher magnetic beads handler (Thermo).

### Single-point phage enzyme-linked immunosorbent assay

For each selection round, the amplified phage pool from the prior round was used as input and pre-cleared with streptavidin paramagnetic beads. Additionally, 1.5 μM of empty nanodiscs were included as competitors in the solution throughout. Individual clones from the final round were isolated, amplified, and screened for binding to MCR-1 via phage enzyme-linked immunosorbent assay (ELISA) as previously described[63,64]. Binding was assessed using either 20 nM MCR-1 or empty nanodiscs immobilized on NeutrAvidin-coated 96-well plates (Nunc).

### Surface antibody cloning, expression, and purification

Clones showing specific phage binding were selected for sequencing. Sanger sequencing was performed at the University of Chicago Comprehensive Cancer Center DNA Sequencing Facility. Unique clones were then subcloned into the surface antibody (sAB) expression vector RH2.2 (a kind gift from S. Sidhu) using the In-Fusion Cloning Kit

(Takara). Successful cloning was verified by DNA sequencing. Subsequently, the sABs were expressed, purified, and their affinities estimated by ELISA, all as described previously[64].

## Single-particle cryo-EM vitrification and data acquisition

The purified MCR-1–Fab complex was concentrated to 3.0 mg mL$^{-1}$ using a 100 kDa concentrator (Amicon). Immediately prior to vitrification, glycyrrhizic acid was added to the sample to a final concentration of 0.01% in order to reduce deleterious effects of the air-water interface. 3 μL of sample was added to a plasma-cleaned (Gatan Solarus) 0.6/1.0-μm (0.6 μm circular holes and a spacing of 1.0 um between the holes) holey gold grid (Quantifoil UltrAuFoil) and blotted using filter paper on one side for 3.5 s using a Vitrobot (Thermo Fisher Scientific) with a blot force of 3 and a wait time of 30 s, before plunging immediately into liquid ethane for vitrification. The plunger was operating at 4 °C with greater than 95% humidity to minimize evaporation and sample degradation. Micrographs were collected using a Titan Krios electron microscope (FEI) at the Columbia University Cryoelectron Microscopy Center, equipped with an energy filter and a K3 direct electron detection filter camera (Gatan K3-BioQuantum) using a 0.83 Å pixel size. An energy filter slit width of 20 eV was used during the collection and was aligned automatically every hour using Leginon[66]. Data collection was performed using a dose of 58.2 e$^-$/Å$^2$ across 50 frames (2.5 s per exposure) at a dose rate of approximately 16.1 e$^-$/pix/s, using a set defocus range of -0.8 to -2.2 μm. A 100 μm objective aperture was used. 5965 micrographs were recorded over a 24 h collection.

## Data processing

Movie frames were aligned using Patch Motion Correction implemented in cryoSPARC v.4.0.1[67] using a B-factor during alignment of 500. Contrast transfer function (CTF) estimation was performed using Patch CTF as implemented in cryoSPARC v.4.0.1. Blob picker in cryoSPARC v.4.0.1 was used to pick particles, and inspect picks was used to curate the picks. This resulted in 1,234,373 particles, which were then subjected to 2D classification in cryoSPARC v.4.0.1. One round of ab initio reconstruction was performed in cryoSPARC v.4.0.1 using three classes, with a maximum resolution set at 7 Å and an initial resolution at 9 Å; the best class from the three-class ab initio was selected, resulting in a stack of 17,848 particles. Further 2D classification was carried out on this particle stack, after which 2109 particles were selected and used as input to manually curate exposures so as to select only those micrographs containing at least one picked particle. The resulting 1618 exposures and particle stack were used as inputs in cryoSPARC v.4.0.1 for Topaz training, which uses deep-learning models to automatically pick particles[68], using an expected number of particles of 100. The Topaz training model was then used in Topaz extract to pick particles among the full set of exposures, resulting in a stack of 383,413 particles, which was then extracted with a 384-pixel size box. Heterogeneous refinement was carried out on this particle stack in cryoSPARC v.4.0.1 using volumes from the ab initio reconstructions along with four decoy classes with a "Batch size per class" of 30,000. This heterogeneous refinement was repeated three times using the particle output associated with the volumes from the previous heterogeneous refinement as input for the next heterogeneous refinement. From this final heterogeneous refinement, particles from the best class were selected (66,623 particles), and the particles were re-extracted and subjected to non-uniform refinement in cryoSPARC v.4.0.1, resulting in a 4.04 Å reconstruction. Using a mask covering MCR-1 and the Fab, local refinement using non-uniform regularization was performed in cryoSPARC v.4.0.1, resulting in a 3.85 Å density map. The particles were further sorted by two class heterogeneous refinement in cryoSPARC v.4.0.1 using the map from the 3.85 Å local refinement and a 10 Å lowpass filtered map as inputs, and using a

"Batch size per class" of 30,000, an initial resolution of 5 Å, and a "Resolution of convergence criteria" of 100. 32,048 particles associated with the lowpass filtered volume were excluded, with the remaining 34,575 particles then subjected to non-uniform refinement, yielding a map with a resolution of 3.73 Å. In an attempt to increase the final number of particles we performed multiple additional rounds of heterogeneous refinement, using the 383,413 particle-stack from Topaz picking as input along with three old ab initio volumes and the 3.73 Å volume. Following this, particles from the best class were selected (46,676 particles). The particles were subjected again to non-uniform refinement and local refinement, generating a 3.73 Å map. The previously described two-class heterogeneous refinement, using a 10 Å lowpass filtered map as one input volume, was performed, yielding 32,882 particles. These were used as input for an additional non-uniform refinement to produce a 3.77 Å density map. Finally, local refinement was performed using the previous mask, resulting in a final resolution of 3.58 Å (Supplementary Fig. 3a).

## Structural model building and refinement

An initial AlphaFold2[44] model of MCR-1 was flexibly fitted and refined into the 3.58 Å map following local refinement using *Namdinator*[43]. Coot[69–71] and PHENIX[72,73] were then iteratively used to further refine the MCR-1 model by manually fitting and adjusting the backbone and most of the side chains into the map. Two extra densities were observed within the map, into which PE (18:1(9Z)/18:1(9Z); PubChem ID 53480897) and KLA (PubChem ID 11355423) were fitted, respectively.

## Model Analysis

Chimera[74] and ChimeraX[75] were used to visualize the structures presented in the figures.

## LC-MS analysis of MCR-1 in nanodisc to identify PE lipids

LC-MS analysis was performed as previously described[76] using a SYNAPT XS mass spectrometer operating in negative ionization mode. Briefly, nanodisc samples of 0.1–3 μL were injected without dilution onto a Waters XBridge C8 Direct Connect HP column (10 μm, 2.1 × 30 mm) for online trapping and desalting, followed by chromatographic separation on a Waters Premier ACQUITY UPLC CSH C18 column (1.7 μm, 2.1 × 100 mm) using a 21-minute gradient. The column was maintained at 40 °C, with a flow rate of 0.25 mL/min. The mobile phase consisted of 60/40 (v/v) acetonitrile/water with 10 mM ammonium acetate (MPA) and 90/10 (v/v) isopropanol/acetonitrile with 10 mM ammonium acetate (MPB).

All data acquisition of full-scan MS1 and FastDDA spectra was collected over a mass range of 50–2000 *m/z*. Electrospray ionization (ESI) was performed with a capillary voltage of -2.0 kV, a sampling cone voltage of 40 V, a source temperature of 120 °C, and a desolvation temperature of 450 °C. Desolvation gas flow was maintained at 700 L/h. The FastDDA method excluded precursor ions within the 50–350 m/z range to minimize interference from potential impurities, applying a mass resolution of 4.7 for low-mass ions and 15 for high-mass ions during MS/MS selection. In each MS scan, seven precursor ions were selected for fragmentation with a ± 100 ppm tolerance per cycle. To enhance fragmentation efficiency, trap MS/MS collision energy ramps were set to 5–10 V for low-mass ions, while high-mass ions were fragmented using collision energies of 75–155 V.

Lipid identification was performed using MS-DIAL (version 5.3) as previously described[77]. Mass tolerances were set to 0.01 Da for MS1 and 0.1 Da for MS2 to maintain high precursor ion accuracy while minimizing the exclusion of fragment ions. Fragmentation spectra were compared against the LipidBlast database to confirm structural assignments. Lipid annotations were validated based on mass error, MS/MS fragmentation coverage of headgroup and fatty acyl fragments.

## Matrix-assisted laser desorption ionization-time of flight (MALDI-TOF) MS

Lipid A from purified MCR-1 protein was isolated as previously described[78]. For matrix-assisted laser desorption ionization-time of flight (MALDI-TOF) mass spectrometry, samples were prepared and analyzed as previously described[78] using the matrix 5-chloro-mercaptobenzothiazole. Spectra were acquired using negative ion reflectron mode (Autoflex Speed mass spectrometer, Bruker Daltonics) with 500 single laser shots averaged for each sample. Sample data were processed using the FlexControl 3.4 and FlexAnalysis 3.4 software from Bruker Daltonics.

## Mutagenesis

Mutations of MCR-1 in the pWSK29 vector[79] were generated with an in-house method using *Thermococcus kodakaraensis* (KOD) polymerase (Novagen) or Q5 Site-Directed Mutagenesis kit (New England Biolabs) and custom primers (Supplementary Table 2).

## Expression test of MCR-1 mutants

Wild-type and all *mcr-1* mutants were cloned into the pWSK29 vector and used to transform *E. coli* K-12 MG1655 competent cells before being grown overnight in 2xYT medium supplemented with 100 μg mL$^{-1}$ carbenicillin at 37 °C with shaking (200 rpm). The next day, 31.5 mL of the same medium was inoculated with the starter culture at a 1:100 ratio and left to grow at 37 °C with shaking (200 rpm) until the $OD_{600nm}$ reached 0.6-0.8. The temperature was then reduced to 22 °C, protein expression was induced with 0.2 mM IPTG, and the culture was incubated overnight with shaking (200 rpm). Cells were collected by centrifugation (1959 x g for 15 min) at 4 °C, washed once with PBS, then centrifuged again to produce a solid pellet. Cell pellets were resuspended in 1.5 mL lysis buffer containing 20 mM HEPES pH 7.5, 200 mM NaCl, 20 mM $MgSO_4$, 10 μg mL$^{-1}$ DNase I, 8 μg mL$^{-1}$ RNase A, 1 mM TCEP, 1 mM PMSF, and Complete Mini EDTA-free protease inhibitor cocktail (Roche) used according to the manufacturer's instructions. Cells were lysed by sonication and the lysate was solubilized for two hours with DDM, added to a final concentration of 1% (w/v). The solubilized material was obtained by centrifugation for 30 min (16,110 x g) at 4 °C, and the supernatant was mixed with Ni-NTA agarose beads with binding allowed to proceed overnight in the presence of 40 mM imidazole. The beads were then loaded on a column and washed with five column volumes of 20 mM HEPES pH 7.5, 500 mM NaCl, 75 mM imidazole, and 0.03% (w/v) DDM. Protein was eluted with two column volumes of 20 mM HEPES pH 7.5, 150 mM NaCl, 300 mM imidazole, and 0.03% (w/v) DDM (100 μL). Imidazole was removed from the eluted protein by exchanging buffer to 20 mM HEPES pH 7.5, 200 mM NaCl, and 0.03% (w/v) DDM by concentrating the protein (Amicon Ultra 0.5 mL; 50 kDa cut-off) and washing the sample with five column volumes of buffer (500 μL) and repeating this five times, subsequently concentrating the samples to 20 μL. The elutes were then separated on a 14% SDS-PAGE gel to confirm expression (Supplementary Fig. 7a).

## Engineering of strains synthesizing 1-dephosphorylated or 4′-dephosphorylated lipid A species

To generate *E. coli* strains producing 1-dephosphorylated lipid A, we chromosomally replaced the native *lpxT* gene with the gene (*lpxE*) encoding the *Francisella* lipid A 1-phosphatase. *lpxE* expression was placed under the control of a constitutive insulated promoter (iP) termed proD by Davis and colleagues[80]. To do this, a sequence of the proD-insulated promoter was first synthesized and subcloned into pUC57 by Genscript. Plasmid pKD4 was modified by a TC insertion at position 1483-bp using primers F1-R1, generating an EcoRI restriction site. The modified plasmid was named pKDE (Supplementary Fig. 15, Supplementary Table 3). The insulated promoter and downstream ribosome binding site (RBS) sequence was amplified using primers F2-

R2 (Supplementary Table 3), digested with EcoRI-NdeI and subcloned into pKDE. The resulting plasmid pKDE-iP containing the region P1-FRT-Kan-FRT-iP-RBS served as a template for subsequent assembly PCR steps (Supplementary Fig. 15).

To chromosomally replace *lpxT* with *lpxE* (Δ*lpxT*::*lpxE*) fragment P1-FRT-Kan-FRT-iP-RBS was amplified using primers F3-R3 (Supplementary Table 3), yielding a 1.6 kb PCR product. A 0.72-kb *lpxE* ORF was amplified from pQLpxE$_{Fn}$[81] with F4-R4 primers (Supplementary Table 3) containing 19 nt at the 5′-end that overlaps iP-RBS at the 3′-end. Both fragments were spliced together by SOEing PCR[82] using primers F3-R4 and generating a ~2.3 kb kan-linked *lpxE* product (P1-FRT-Kan-FRT-iP-RBS-lpxE) (Supplementary Fig. 15).

The kan-linked *lpxE* was then amplified using primers F5-R5 (Supplementary Table 3) containing the 5′ end 50-nt flanking *lpxT* homologous regions (Supplementary Fig. 15). DY330 strain was transformed with the 2.4 kb PCR product allowing chromosomal homologous recombination based on λ Red recombinase system[83]. Candidate recombinants were selected on 30 μg/ml kanamycin. The Δ*lpxT*::*lpxE* kan$^R$ was integrated into W3110 chromosome by P1*vir* phage transduction and the gene resistant cassette excised using the FLP recombinase system encoded by plasmid pCP20[84]. The cured and marker-less W3110 Δ*lpxT*::*lpxE* strain was transduced with Δ*eptA*::kan P1 phage generating W3110 Δ*lpxT*::*lpxE*, Δ*eptA*.

For generating an *E. coli* strain lacking the 4′-phosphate group, we used the *Francisella* 4′-phosphatase LpxF. Since LpxF only functions on penta-acylated lipid A we chose to replace the native *lpxM* gene with *lpxF*. *lpxM* encodes the last acyltransferase for lipid A synthesis and its deletion results in penta-acylated lipid A production with little to no impact on growth. To do this, the 0.6-kb *lpxF* ORF was amplified from pQLpxF$_{Fn}$[81] using primers F6-R6 (Supplementary Table 3). The P1-FRT-Kan-FRT-iP-RBS fragment from pKDE-iP was fused to *lpxF* by 22-nt overlapping region using SOEing PCR and primers F3 and R6. The generated 2.2-kb kan-linked *lpxF* product (P1-FRT-Kan-FRT-iP-RBS-lpxF) served as template for primers F7-R7 (Supplementary Table 3), which contain the 5′-ends 50-nt flanking *lpxM* homologous regions (Supplementary Fig. 15). After transformation and chromosomal recombination in DY330, Δ*lpxM*::*lpxE* kan$^R$ was transferred into BN1 strain by P1 phage transduction. All deletions were verified by PCR of target genomic DNA and Sanger sequencing. All strains and plasmids used for this study are listed in Supplementary Table 3.

## Functional analysis of MCR-1 variants by TLC

Overnight cultures were diluted to $OD_{600}$ ~ 0.05 in LB containing 2.5 μCi/ml $^{32}P_i$ (Revvity) and antibiotics and IPTG added where appropriate. Cells were harvested at $OD_{600}$ ~ 1.0. Lipid A was isolated as previously described via mild-acid hydrolysis (pH 4.5, 100 °C) that cleaves the linkage between lipid A and the first Kdo sugar[78]. The lipid was then isolated via Bligh-Dyer extraction[78]. Radiolabeled lipids were visualized by phosphorimaging analysis using Amersham Typhoon Biomolecular Imager (Cytiva).

## Polymyxin minimal inhibitory concentration (MIC) determination

MICs were determined by E-strip (BioMérieux). Overnight cultures of *E. coli* W3110 (Δ*lpxT*, Δ*eptA*) expressing MCR-1 variants were diluted 1:100 into fresh LB medium. Cultures were grown to mid-log phase, back diluted 10-fold, and spread onto LB plates. After drying, sterile gradient polymyxin E-strips were applied and the plates incubated overnight at 37 °C. The MIC was assigned as the value where the zone of growth inhibition intersects with the E-strip. Where appropriate, antibiotics and IPTG were included in the liquid growth media and agar plates.

## Efficiency of plating assay

Cultures grown at 37 °C overnight were normalized to $OD_{600}$ 1.0 and serially diluted 1:10 in LB broth. Dilutions were transferred onto LB

agar plates containing antibiotics using a replica plater (Sigma-Aldrich). Cells were incubated for 16 h at 37 °C.

## Statistics and reproducibility

For reproducibility, all SDS-PAGE gels were run in duplicate (Supplementary Fig. 1e, Supplementary Fig. 7a), and TLC data are representative of three biological replicates.

## MD simulations

The cryo-EM structure was used to build molecular models for MCR-1 in the presence and absence of substrates for State 1, while an AlphaFold2 model was used for State 2. The MemProtMD pipeline[85,86] was used to permit the assembly and equilibration of a POPG:POPE bilayer around MCR-1 in both conformations, without its substrates. The input protein was aligned according to the plane of the membrane with MEMEMBED[87] and converted to a CG representation using the Martini 3 force field[88]. Intra-chain elastic network with a force constant of 500 kJ mol$^{-1}$ nm$^{-2}$ was used to restrain the secondary structure of the protein. A POPG:POPE bilayer at 1:4 molar ratio was built around the protein using the insane protocol[89]. The system was solvated with Martini 3 waters and NaCl was added in a concentration of 150 mM to neutralize the systems. The system was minimized with the steepest descents method, followed by 1 μs MD simulation with 20 fs time step, in the NPT ensemble with velocity rescaling thermostat and semi-isotropic c-rescale pressure coupling. The final snapshot was converted back to atomic details using CG2AT[90] with the "align" method to retain the coordinates of the MCR-1 EM structure during conversion. The substrates PE and KLA were added to the systems using the coordinates from the cryo-EM structure. The *Nm*EptA crystallographic structure (PDB ID: 5FGN[16]) was used to model the position of Zn$^{2+}$ for the simulations starting from the State 1 conformation. According to the two-step reaction mechanism, the second reaction requires two Zn$^{2+}$ ions[12]. The initial position of the second Zn$^{2+}$ was taken from the di-Zn$^{2+}$ MCR-1 crystallographic structure (PDB ID: 5LRM[28]). The complete systems were further equilibrated for 10 ns maintaining the structure of the protein and ligands restrained. Five repeats of unrestrained 500 ns MD simulations were performed for each system. Atomistic simulations were performed using the CHARMM36m force field[91] with the TIP3P water model. The protonation state of residues was assigned by assuming standard protonation based on canonical pKa values at pH 7.0. All simulations were performed in the isothermal-isobaric ensemble at 310 K and 1 bar using a time-step of 2 fs. Pressure was maintained at 1 bar with a semi-isotropic compressibility of $4 \times 10^{-5}$ ($\tau_P = 1 ps$) using the Parrinello-Rahman barostat[92]. Temperature was controlled using the velocity-rescale thermostat[93] ($\tau_T = 0.1 ps$), with the solvent, lipids and protein coupled to an external bath. The long-range electrostatic interactions were computed with the Particle Mesh Ewald method[94], with a cut-off of 1.2 nm. Van der Waals interactions were computed using a force-switch modifier between 1.0 and 1.2 nm. All MD simulations were performed using GROMACS 2022[95–97] and analyzed using GROMACS tools and MDAnalysis[98,99].

## Reporting summary

Further information on research design is available in the Nature Portfolio Reporting Summary linked to this article.

# Data availability

The Cryo-EM density map has been deposited into the Electron Microscopy Data Bank (EMDB) under accession code EMD-49896. The model has been deposited in the Protein Data Bank (PDB) under accession code 9NWW. The MD simulation data generated in this study are available as a supplementary dataset on Zenodo (https://doi.org/10.5281/zenodo.17258531). All raw gels and full TLC plate images are available in the Supplementary Information. Source data are provided with this paper.

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

## Acknowledgements

The authors thank Anne-Catrin Uhlemann and members of the Mancia laboratory for their insights and helpful discussions. We would also like to thank the Columbia University cryo-EM facility. This work was funded by NIH grants GM132120 (to F.M.); AI74416, AI176776, and AI150098 (to M.S.T.); and AI181556 (to A.P.Z.). Research in the laboratory of P.J.S. was funded by Wellcome (208361/Z/17/Z), MRC, BBSRC, EPSRC, NIH, JPIAMR and the Howard Dalton Centre. This project made use of time on ARCHER2 granted via the UK High-End Computing Consortium for Biomolecular Simulation, HECBioSim (http://www.hecbiosim.ac.uk, supported by EPSRC (grant no. EP/R029407/1). P.J.S. and M.B.-B. acknowledge Sulis at HPC Midlands +, which was funded by the EPSRC on grant EP/T022108/1, and the University of Warwick Scientific

Computing Research Technology Platform for computational access. Some of this work was performed at the New York Structural Biology Center.

## Author contributions

A.P.Z., with help from K.U.A., performed the protein expression and purification. S.K.E., K.N., and A.A.K. identified and purified the Fabs. A.P.Z. produced and analyzed the cryo-EM data with help from R.N. and R.J.C., and built the model with help from R.N. Mutations were cloned by A.P.Z., B.K., C.M.H., and M.S.T., and TLC analyses to assess MCR-1 function were carried out by C.M.H. and M.S.T. LC-MS was performed by G.Z. and M.T.M, and MALDI-TOF MS was performed by C.M.H. and M.S.T. Engineering of cell strains to study site-selective phosphate modification of lipid A was conducted by C.M.H. and M.S.T. All MD simulations were performed and analyzed by M.B.-B. and P.J.S. A.P.Z., F.M., P.J.S., M.B.-B., C.M.H., and M.S.T. designed experiments and wrote the paper with R.N. Oversight for the entire project was provided by F.M.

## Competing interests

The authors declare no competing interests.
