## [Transparent Peer Review file · Nature Communications]

Mechanistic basis of antimicrobial resistance mediated by the phosphoethanolamine transferase MCR-1

Corresponding Author: Professor Filippo Mancía

Version 0:

Reviewer comments:

Reviewer #1

(Remarks to the Author)

The manuscript entitled "Mechanistic basis of antimicrobial resistance by the phosphoethanolamine transferase MCR-1" by Allen P. Zinkle et al. mainly focuses on cryo-EM-based structural characterization of MCR-1, which is a PEtN transferase. EptA from *E. coli* selectively modifies the 1-phosphate of lipid-A, while MCR-1 shows preferential selectivity towards the 4'-phosphate. In this study, the authors used antigen-binding fragment technology to develop different Fab fragments to increase the overall molecular weight of full-length MCR-1 and employed single-particle cryo-electron microscopy (cryo-EM) coupled with Fab-bound conformation to determine the structure of full-length MCR-1; not only that, the authors identified both PE and lipid-A bound structures, which showed two distinct ligand-binding sites of MCR-1. The authors performed structural analysis, computational studies, cell survival assays, drug-resistance experiments, and genetic and biochemical analyses. They proposed a mechanism of action of MCR-1, which involves a significant conformational change during PEtN transfer. They also claimed that their computational studies highlighted how phosphor form transferases, essential for cell envelope synthesis, utilize PEtN or glycerophospholipids as donor substrates.

The authors had performed a considerable amount of work. Also, this manuscript is written okay, and overall, the representation is okay. The authors have resolved the structure of the small protein MCR-1 at high resolution. The overall cryo-EM data processing pipeline and methods section are represented very complicated way. However, more clarification is required for the data representation. Figure rearranging is absolutely important. It is very difficult for the reader to go back and forth between Figure 4, Figure 1, then Figure 2, and so on to correlate the figures and text. There are some flaws, which are described below.

My comments are presented below:

1. Over the past decade, several groups have worked on this MCR-1 protein from *E. coli* and resolved the crystal structure of the catalytic domain of MCR-1 (MCR-1-ED) from *Escherichia coli* (*E. coli*). In that manuscript, the authors identified β - α - β motifs that adopted a "sandwich" conformation and demonstrated two functional states of MCR-1 depending on the physiological conditions. Furthermore, MCR-1 is similar to other known pEtN transferases as reported in *Sci Rep* 6, 38793 (2016) "a phosphoethanolamine transferase for Colistin Resistance" (<https://doi.org/10.1038/srep38793>). Additionally, *BMC Biol* 14, 81 (2016) reported that cMCR-1 is a globular protein with an overall hemispherical shape and a centrally located β -sheet composed of seven β -strands sandwiched between α -helical structures (<https://doi.org/10.1186/s12915-016-0303-0>). Based on these available information, it is essential to describe from what perspective this current study is unique and novel compared to all these published structures. At the same time, the full-length structure is a novel or new finding, which alone is not sufficient. Also, this current structure is resolved with a Fab fragment, and small conformational changes might be possible due to Fab binding; the authors should discuss this in detail how this is affecting the current structure and how is it different from the published structures.
2. Why does the MCR-1 with ND SEC profile peak look so heterogeneous, although incorporated into a nanodisc with Fab (MR6) bound (green) appear more homogeneous? Any reason behind this?
3. The authors mentioned that they have screened seven high-affinity Fab candidates (MR1-MR7) for complex formation with MCR-1 (Extended Data Fig. 2a), and that MR6 was selected due to its high binding affinity (Extended Data Fig. 2b, c). However, from Extended Data Fig. 2b, c, I cannot identify how the authors selected strong complex formation. Multi-point sAB ELISA: EC50 estimation value showed that MR5 and MR6 demonstrated better affinity, whereas the phage display Fab library showed that MR5 and MR1 were almost equivalent to MR6.
4. Extended Data Fig. 3, Cryo-EM analysis of the MCR-1–Fab complex in nanodisc, is presented in such a way that it is very difficult to follow. How the authors processed the data is unclear. There is a high chance of duplication of raw data images because the same 66,623 particles were used twice for heterogeneous, non-uniform, and local refinement.

5. MCR-1 closely resembles the NmEptA crystal structure, with the greatest deviation occurring along the BH region—what is the BH region? From the figure, it is not very clear what the actual difference between MCR-1 and NmEptA crystal structures is.
6. I did not understand the conformational flexibility of the MCR-1 periplasmic domain from the figure. The authors claimed that the NmEptA crystal structure (State 1), and the other two predictions display an alternative conformation (State 2). However, I am unable to visualize the different states. If there is conformational flexibility, why is it not visible in the 3D classification? Additionally, in 2D averages, the TM is not clearly visible. The authors achieved high-resolution structures, and I think in that case, TM should be visible in 2D.
7. The author should show the map-to-model fitting, angular distribution, and 3D FSC both with and without a mask.
8. The FSC curve of this reconstruction looks very poor. Any particular reason for this?

Reviewer #2

(Remarks to the Author)

This manuscript reveals a cryo-EM structure, functional and computational data to improve our understanding of the MCR-1 mechanism. MCR-1 is an enzyme directly responsible for polymyxin resistance, by adding PEtN group onto lipid A, the target of polymyxin. Overall, this manuscript is clear, timely, and contributes important knowledge to the field on how MCR-1 (and similar enzymes) function. It combines really multi-disciplinary and complementary techniques to present a convincing two step model of MCR-1 mechanism. The figures and text are clear throughout.

Below are some suggestions to improve the manuscript quality/clarity in a few places.

One less minor comment:

-Line 299 - The authors should confirm their *in silico* mutagenesis with their wetlab experiments (TLCs, polymyxin B data). Unless these mutants are unstable in which case this should be stated.

Minor comments:

- Line 34 - "Unprecedented" may be overselling slightly
- Line 99 - add "fab" as abbreviation here
- Line 113 - "Two step procedure" is a bit vague and out of place. Perhaps the extended figure 1 should be referenced here.
- Line 133 - some justification for why just POPG nanodiscs were used would be good. Because POPE is the substrate?
- Line 137 - Justification for picking MR6 - was it actually the best when you consider background in elisa for buffer/empty nanodisc? Clearly it worked in the end, but the whole reason for selection of that one over others isn't super clear
- Line 149 - ref Fig 1C here (rather than after a few sentences)
- Line 154 - Can the authors confidently conclude zinc isn't present? When you fit the previous EPTA structure (5FGN) into the cryo map, there is density present at the expected Zn binding site, so not sure if a side chain has been modelled into that density instead? Do the authors have any biochemical evidence to suggest their purified protein doesn't bind zinc, or if this is just because they can't unambiguously assign it given the resolution?
- Line 162-164 - sentence a little clunky, split into 2?
- Line 175 - refer to N108 and T112 having diminished activity and refer to numbers in Extended data fig 7b. Are these numbers from replicates? I think this needs to be made clear if any conclusions are being made, especially when these residues are particularly marginal
- Line 198 - D102A and D102A/R190A mutants seem to be purified at lower levels than others, so perhaps has lower stability? Can the authors also justify why they did purifications rather than western blots to look at cellular accumulation, as all subsequent assays are done in the cell? Is the tagged version not functional?
- Line 212- Can the authors justify why they only tested some mutants in the polymyxin assay and why those particular ones were chosen?
- Line 283 - Why are the non-lipid A species in the TLC? And why are they only in the 4' TLC?

Throughout - authors alternate between single and three letter abbreviations for amino acids - would be good to pick one!

Reviewer #3

(Remarks to the Author)

This is an extremely interesting and thorough structural investigation examining the mechanism of action of phosphoethanolamine transferase MCR-1 which catalyses the transfer of PEtN from a donor PE lipid bound to one site to modify Lipid A, increasing its overall charge. The authors use a combination of cryo-EM, biochemical mutagenesis, and molecular dynamics simulations to characterise a substrate-bound conformation of MCR-1. The cryoEM density maps show clear densities in two putative binding sites, one for the donor PE lipid, located ~20Å from the acceptor Lipid A site. The resolution was sufficient to allow both PE lipid and Kdo2-Lipid A to be unambiguously modelled into the second site. Mutations of key residues involved in the lipid A binding site showed a decrease in Lipid A modification, confirming the Lipid A binding site. Molecular dynamics simulations identified a major conformational change between the PE binding domain and the transmembrane domain, providing a novel mechanism for PEtN transfer to Lipid A.

This work is of significance across the fields of structural biology and microbiology, as it shows a previously unidentified mechanism for PEtN transferases that may be targeted for future antimicrobial design, and increases our general understanding of bacterial cellular processes.

The manuscript is clearly written and logically set out. The cryoEM, biochemical analysis and MD simulations supported each other. The results flowed and logically from one point to the next and supported the conclusions that were drawn in the manuscript.

Overall, I had a few minor comments that should be addressed:

- The authors describe State 1 and State 2 from the MD simulations, with State 1 corresponding to the cryoEM structure. They note that State 1 is consistent with the previously solved crystal structure of NmEptA identified by Anandan et al in 2017. They state that the NmEptA structure is apo, but in fact this is not strictly true, as there is a DDM detergent molecule bound in the PE binding site. Density corresponding to a Zn ion was also noted in the NmEptA structure.
- Anandan et al performed MD simulations on NmEptA and also noted a “tumble weed” type conformational change between the transmembrane domain and the catalytic domain. How did this second conformation captured by Anandan et al correspond to State 2 identified in the MD simulations presented here? From a quick inspection of the supporting information of Anandan et al, the two conformations appear quite different.
- In terms of the MD simulations, it would be good to know which residues coordinated the Zn ions, and how stably Zn bound to the binding site of the protein. For what proportion of the simulation did Zn remain stably coordinated by the proposed Zn binding residues? Both the first and second Zn ions were placed in the respective binding sites based on crystallographic information from other proteins.
- The methods describe an unusual set-up for the simulations. If I am understanding correctly, the authors first coarse-grained the structure and used the Martini 3 force field, applied an elastic network (with a high force constant) to the backbone, built a CG PE/PG membrane around the protein, ran 1 μ s of CG simulations, then backmapped it to atomistic resolution, then ran replicate 500 ns atomistic simulations under different conditions and analysed these. This seems like an awfully convoluted way to set up a simulation. Why was this done? Why not just take the cryoEM structure and equilibrate it in an atomistic membrane simulations? What purpose do the CG simulations serve, except to cite a specific set of tools for CG membrane generation and back mapping?
- There are also gaps in the computational methodology that prevent reproducibility. For example, which Martini water was used? I'm assuming Martini 3 water parameters, but this should be explicitly stated. The RMSD between the back mapped and cryo-EM structure should also be given. I would expect this to be very low given the high force constant that has been used in the elastic network.
- Restraints were used in the equilibration period of the atomistic simulations. What was the relaxation protocol for the restraints? How was the system equilibrated?
- Molecular dynamics checklist point 4c asks. “Are other parameters for the system setup described in the text, such as protonation state, type of structural restraints if applied, nonbonded cutoff, thermostat and barostat, etc.?” These have not been included in the manuscript. Please explicitly state what force field, water model and ion model was used for the atomistic simulations, and how the protonation state was assigned. There is no reference to the atomistic force field in either the methods or in Extended Data Table 5. Likewise, there is no reference to the pressure coupling or temperature coupling TauP or TauT, or to the cut-off method for the nonbonded interactions. Without this information, the simulation protocol cannot be reproduced.

Version 1:

Reviewer comments:

Reviewer #1

(Remarks to the Author)

The authors have significantly revised the manuscript and addressed all the major concerns. They have modified some figures and have responded to most of the reviewers' suggestions appropriately. Their responses are convincing.

I have one minor suggestion and one comment:

Minor suggestion:

Extended Data Fig. 3 is improved but still not very clear. I had to read it multiple times to fully understand what is shown. In the rebuttal, the authors explained: “we did not re-use these particles, but rather used the new volume (3.73 Å) as an input volume for a new round of heterogeneous refinement using the original 383,413 particle stack and replacing the original input volume representing MCR-1 in nanodisc with Fab (shown in green).” This should also be mentioned in the figure legend, with the new 3.73 Å volume marked as the initial model for the new round of heterogeneous refinement. While this is clearly described in the Methods section, adding it to the figure legend and marking it in the figure would improve clarity.

Comment:

Regarding Reviewer #1, Question 6: the authors provided an additional set of eight 2D class averages showing the transmembrane domain (circled in red). However, the TM region is still not very clear. Many cryo-EM papers on GPCRs, transporters, and efflux pumps published over the last 5–10 years show transmembrane regions in detergent, liposomes, or lipids more clearly than in the present case. That said, this does not affect the 3D reconstruction, the overall findings, or the conclusions of the manuscript.

I have no further questions or comments. This manuscript is suitable for publication.

Reviewer #2

(Remarks to the Author)

The manuscript remains timely and novel. My positive comments about the manuscript from the first round of review remain: "This manuscript reveals a cryo-EM structure, functional and computational data to improve our understanding of the MCR-1 mechanism. MCR-1 is an enzyme directly responsible for polymyxin resistance, by adding PEtN group onto lipid A, the target of polymyxin. Overall, this manuscript is clear, timely, and contributes important knowledge to the field on how MCR-1 (and similar enzymes) function. It combines really multi-disciplinary and complementary techniques to present a convincing two step model of MCR-1 mechanism. The figures and text are clear throughout."

The below comments are specifically in response to the authors response.

The authors seemed reluctant to partake in any of the minor lab-based experiments suggested for this paper. The justification for not doing western blots was fine, but still it seems like a fairly quick and simple experiment to do that would have been insightful. Furthermore, the justification for selecting only some of the mutants in their experiments was not that clear, and in some cases not explained, when again it should have been fairly simple to add on.

I don't think doing these additional experiments are essential for the paper to be published, but it would have been an improvement for fairly little hands on time.

All of the easier, text-based, edits were addressed well.

Reviewer #3

(Remarks to the Author)

The authors have adequately addressed my queries in their revised manuscript.

Response to reviewers

Zinkle et al. “Mechanistic basis of antimicrobial resistance by the phosphoethanolamine transferase MCR-1”

We thank the Reviewers for their helpful comments and critical assessment of our manuscript. Below is our point-by-point response to the comments from Reviewers #1, #2, and #3:

Reviewer #1:

The manuscript entitled "Mechanistic basis of antimicrobial resistance by the phosphoethanolamine transferase MCR-1" by Allen P. Zinkle et al. mainly focuses on cryo-EM-based structural characterization of MCR-1, which is a PEtN transferase. EptA from E. coli selectively modifies the 1-phosphate of lipid-A, while MCR-1 shows preferential selectivity towards the 4'-phosphate. In this study, the authors used antigen-binding fragment technology to develop different Fab fragments to increase the overall molecular weight of full-length MCR-1 and employed single-particle cryo-electron microscopy (cryo-EM) coupled with Fab-bound conformation to determine the structure of full-length MCR-1; not only that, the authors identified both PE and lipid-A bound structures, which showed two distinct ligand-binding sites of MCR-1. The authors performed structural analysis, computational studies, cell survival assays, drug-resistance experiments, and genetic and biochemical analyses. They proposed a mechanism of action of MCR-1, which involves a significant conformational change during PEtN transfer. They also claimed that their computational studies highlighted how phosphor form transferases, essential for cell envelope synthesis, utilize PEtN or glycerophospholipids as donor substrates. The authors had performed a considerable amount of work. Also, this manuscript is written okay, and overall, the representation is okay. The authors have resolved the structure of the small protein MCR-1 at high resolution. The overall cryo-EM data processing pipeline and methods section are represented very complicated way. However, more clarification is required for the data representation. Figure rearranging is absolutely important. It is very difficult for the reader to go back and forth between Figure 4, Figure 1, then Figure 2, and so on to correlate the figures and text. There are some flaws, which are described below.

We thank the Reviewer for their supportive comments. The figures are referenced in the order they are referenced in the text. After careful consideration, we were unable to find a rearrangement of the panels within or between Figures 1, 2, and 4 that was satisfactory with the narrative from the text. We understand that it is not ideal to refer back to Fig. 2 through specific in-text references late in the results and discussion sections, respectively (lines 343 and 395-397, for example). However, we have reviewed the figure references and believe they are all relevant and help the reader understand the details of the text. We have edited our cryo-EM data processing pipeline figure (Extended Data Fig. 3a) and the methods section for clarity, which we describe later in greater detail. Please see our responses to the individual comments below.

- 1. Over the past decade, several groups have worked on this MCR-1 protein from E coli and resolved the crystal structure of the catalytic domain of MCR-1 (MCR-1-ED) from Escherichia coli (E. coli). In that manuscript, the authors identified β - α - β - α motifs that adopted a "sandwich" conformation and demonstrated two functional states of MCR-1 depending on the physiological conditions. Furthermore, MCR-1 is similar to other known pEtN transferases as reported in Sci Rep 6, 38793 (2016) "a phosphoethanolamine transferase for Colistin Resistance" (<https://doi.org/10.1038/srep38793>). Additionally, BMC Biol 14, 81 (2016) reported that cMCR-1 is a globular protein with an overall hemispherical shape and a centrally located β -sheet composed of seven β -strands sandwiched between α -helical structures (<https://doi.org/10.1186/s12915-016-0303-0>). Based on these available*

information, it is essential to describe from what perspective this current study is unique and novel compared to all these published structures. At the same time, the full-length structure is a novel or new finding, which alone is not sufficient. Also, this current structure is resolved with a Fab fragment, and small conformational changes might be possible due to Fab binding; the authors should discuss this in detail how this is affecting the current structure and how is it different from the published structures.

We agree it is important to emphasize the novelty of this study compared to other structural studies of PEtN transferases, particularly X-ray crystallographic studies of the periplasmic domain (PD). Accordingly, we have included additional sentences in the introduction (lines 75-79) and discussion (lines 354-355) sections: the former describing the structural motifs present in the PD, as well as further highlighting the two distinct and physiologically relevant states observed within the PD (with or without additional density present near the side chain of the active site Thr285) and including an additional citation (<https://doi.org/10.1038/srep38793>; now reference 23); and the latter underscoring how our cryo-EM structure is the first to structurally define the transmembrane region of an MCR protein, as well as the first to identify the lipid A binding site (e.g., is lipid A-bound) (lines 354-355). We agree that the Fab binding could induce small conformational changes in the protein, and as such we have included additional commentary on this possibility in the discussion section (lines 382-388), specifically comparing the region where MR6 binds on MCR-1 to the same region in the full-length EptA crystal structure. Indeed, comparison of our Fab-bound cryo-EM structure with the crystal structure of full-length *NmEptA* shows no significant difference in conformation along the MR6–MCR-1 binding interface compared to that same region in *NmEptA*, so we have reason to believe that MR6 does not induce large conformational changes.

- 2. Why does the MCR-1 with ND SEC profile peak look so heterogeneous, although incorporated into a nanodisc with Fab (MR6) bound (green) appear more homogeneous? Any reason behind this?*

In Extended Data Fig. 1d, the elution profile for MCR-1 in nanodisc (orange) appears more heterogeneous than that of MCR-1 in DDM (blue) or in nanodisc with Fab (MR6) (green). In our opinion and experience, this is because there is most often some amount of protein, including both MCR-1 and the membrane scaffold protein (in this case, MSP1D1), that forms aggregates during the nanodisc reconstitution process. These aggregation peaks (with peak fractions eluting at ~8.75 mL and 10.0 mL, respectively) elute prior to non-aggregated MCR-1 in nanodisc (with peak fraction eluting ~11.0-11.5 mL), resulting in a more heterogeneous profile. From measurements taken using a REFERYN mass photometer (data not shown), the molecular masses of these aggregation peaks are outside the range typical for MCR-1 in MSP1D1 nanodiscs (~170-230 kDa). The fraction representative of the peak eluting at ~11.0-11.5 mL was collected and incubated with Fab, thereby limiting aggregation in the final elution profile (that of MCR-1 in nanodisc with MR6).

- 3. The authors mentioned that they have screened seven high-affinity Fab candidates (MR1-MR7) for complex formation with MCR-1 (Extended Data Fig. 2a), and that MR6 was selected due to its high binding affinity (Extended Data Fig. 2b, c). However, from Extended Data Fig. 2b, c, I cannot identify how the authors selected strong complex formation. Multi-point sAB ELISA: EC50 estimation value showed that MR5 and MR6 demonstrated better affinity, whereas the phage display Fab library showed that MR5 and MR1 were almost equivalent to MR6.*

We tested multiple Fabs during cryo-EM screening, including MR1 and MR5, but found a greater incidence of Fab dissociation and preferred orientation biasing in the data compared to data from sample incubated with MR6. Oftentimes, the highest affinity Fab is not necessarily the one that produces the best results for cryo-EM. While we agree that our ELISA

data in Extended Data Fig. 2b, c indeed show that MR6 is not the highest affinity Fab among MR1-MR7, it nonetheless is still a top binder along with MR1 and MR5. To avoid confusion, we have revised the text, beginning at line 144, to now state "... MR6 was identified as a top binder along with MR1 and MR5 (Extended Data Fig. 2b, c). However, MR6 was ultimately chosen based on two-dimensional (2D) class averages obtained from preliminary cryo-EM screening."

4. *Extended Data Fig. 3, Cryo-EM analysis of the MCR-1–Fab complex in nanodisc, is presented in such a way that it is very difficult to follow. How the authors processed the data is unclear. There is a high chance of duplication of raw data images because the same 66,623 particles were used twice for heterogeneous, non-uniform, and local refinement.*

We thank the Reviewer for this feedback, and we agree with the data processing, particularly in Extended Data Fig. 3a, could be improved for clarity, which we have done. We have also similarly revised the data processing methods section (lines 502-538). The 66,623 particle-stack shown in Extended Data Fig. 3a was an output following multiple rounds of heterogeneous refinement after initial particle picking and extraction. These particles were then used for non-uniform and local refinement. Following this, lower quality particles were further sorted out of the 66,623 particle-stack by performing another round of heterogeneous refinement. This was achieved by first applying a lowpass filter (10 Å) to the non-uniform refinement volume (4.04 Å) associated with the 66,623 particle-stack, producing a lowpass filtered volume. We then used this lowpass filtered volume and the original refined volume as the input volumes for heterogeneous refinement with the 66,623 particle-stack, yielding two output particle-stacks – one consisting of 32,048 particles associated with the volume with the lowpass filter applied, and the other consisting of 34,575 particles. This 34,575 particle-stack was then used as input for another non-uniform refinement job, yielding a map at 3.73 Å global resolution. Some confusion likely arises with how we proceeded from here: in the figure, one may conclude that we used the 34,575 particle-stack to perform heterogeneous refinement, yielding a larger particle-stack (46,676 particles). However – and we believe has been clarified in our revised Extended Data Fig. 3a – we did not re-use these particles, but rather used the new volume (3.73 Å) as an input volume for a new round of heterogeneous refinement using the original 383,413 particle-stack and replacing the original input volume representing MCR-1 in nanodisc with Fab (shown in green). Following multiple rounds of heterogeneous refinement using these volumes as input, we were left with a 46,676 particle-stack. Non-uniform and local refinement jobs were performed, and a lowpass filter (10 Å) was applied to the non-uniform refinement volume (3.84 Å). We then used this lowpass filtered volume and the original refined volume as the input volumes for another heterogeneous refinement job with the 46,676 particle-stack, yielding two output particle-stacks, one consisting of 13,794 particles associated with the volume with the lowpass filter applied, and the other consisting of 32,882 particles. This 32,882 particle-stack was then used as input for another non-uniform refinement, yielding a map at 3.77 Å global resolution, and local refinement to produce the 3.58 Å map.

5. *MCR-1 closely resembles the NmEptA crystal structure, with the greatest deviation occurring along the BH region—what is the BH region? From the figure, it is not very clear what the actual difference between MCR-1 and NmEptA crystal structures is.*

The bridging helix (BH) for PEtN transferases is defined as the helix peripheral to the membrane that connects the transmembrane domain with the periplasmic domain (lines 86-87). Given that the EptA crystal structure and our MCR-1 cryo-EM model have very similar overall conformations and differ mainly in the composition of the BH, we agree with the Reviewer that more should be added to describe this region and how it differs between the two available structures. Accordingly, we have referenced panel b from Extended Data Fig. 5 in line 158 along with panel a, which shows the sequence variability in the BH region between

MCR-1 and several EptA homologs. In addition, we have added extra sentences in the results section (lines 156-158), which is in addition to our commentary in the discussion (lines 394-402) which further describe the sequence variation in the BH and speculate on the significance of this.

6. *I did not understand the conformational flexibility of the MCR-1 periplasmic domain from the figure. The authors claimed that the NmEptA crystal structure (State 1), and the other two predictions display an alternative conformation (State 2). However, I am unable to visualize the different states. If there is conformational flexibility, why is it not visible in the 3D classification? Additionally, in 2D averages, the TM is not clearly visible. The authors achieved high-resolution structures, and I think in that case, TM should be visible in 2D.*

The NmEptA crystal structure is in a very similar conformation as that of our MCR-1 cryo-EM structure – we refer to this conformation as “State 1”. Using AlphaFold, two conformations are predicted for P_{ET}N transferases: one, which appears as “State 1” like the experimentally derived structures; and an alternative conformation with the PD of the enzyme rotated to be placed above the opposite side of the transmembrane domain (which we call “State 2”). We visualize the two states in Fig. 3a, where we superimpose the “State 2” model (with the PD colored light blue) – predicted by AlphaFold – onto the transmembrane domain of our cryo-EM structure (with the PD colored light green, as in Fig. 1). We explain in lines 379-382 in the discussion that the use of the MR6 Fab in our cryo-EM data likely precludes the possibility of capturing particles in the “State 2” conformation, as the Fab is bound where the PD domain is predicted to reside in “State 2” (please see Fig. 1b and compare to Fig. 3a). Hence, we were unable to parse different states through 3D classification. In Extended Data Fig. 3b, the transmembrane domain secondary structure is visible in four of the six 2D classes shown (top left, middle right, bottom left, and right). It can be seen located within the nanodisc, between the periplasmic domain and MR6 Fab, which are extending outside the nanodisc. It is possible that the figures were supplied in a resolution that is not ideal for visualizing this, and we are happy to provide an additional eight 2D class averages below that should similarly display the transmembrane domain (circled red).

7. *The author should show the map-to-model fitting, angular distribution, and 3D FSC both with and without a mask.*

We are happy to provide below a map-to-model fitting (panel a), angular distribution (panel b), and 3D FSC curve (panel c) for a non-uniform refinement volume without a mask.

8. *The FSC curve of this reconstruction looks very poor. Any particular reason for this?*

In our opinion, this FSC curve is typical for a small membrane protein, and is the best curve we could achieve with the data we have. Nonetheless, the quality of the density is high.

Reviewer #2:

This manuscript reveals a cryo-EM structure, functional and computational data to improve our understanding of the MCR-1 mechanism. MCR-1 is an enzyme directly responsible for polymyxin resistance, by adding PEtN group onto lipid A, the target of polymyxin. Overall, this manuscript is clear, timely, and contributes important knowledge to the field on how MCR-1 (and similar enzymes) function. It combines really multi-disciplinary and complementary techniques to present a convincing two step model of MCR-1 mechanism. The figures and text are clear throughout.

Below are some suggestions to improve the manuscript quality/clarity in a few places.

1. *Line 299 - The authors should confirm their in silico mutagenesis with their wetlab experiments (TLCs, polymyxin B data). Unless these mutants are unstable in which case this should be stated.*

Of the three mutants tested *in silico*, two (D119A and K401A) were cloned and tested using TLC, as with the others (please see the unpublished TLC data below, with the lanes corresponding to D119A and K401A bordered in red). We did not try the third mutant (R402A).

W3110 DlpXT DeptA
pWSK29::MCR-1

2. Line 34 - “Unprecedented” may be overselling slightly

We have removed the “unprecedented” from line 34 and replaced with “previously undescribed”.

3. Line 99 - add “fab” as abbreviation here

This has been corrected, now in line 104.

4. Line 113 - “Two step procedure” is a bit vague and out of place. Perhaps the extended figure 1 should be referenced here.

We have removed “following a two-step procedure” and revised line 117 to instead state “using metal affinity chromatography (see Methods for more details)”.

5. Line 133 - some justification for why just POPG nanodiscs were used would be good. Because POPE is the substrate?

We have revised the text beginning at line 136 to now read “From a reconstitution screen comparing the elution profiles of purified MCR-1 incorporated into each of six distinct nanodiscs to that of MCR-1 in DDM, nanodisc composed of... (POPG) lipid was chosen, due in part to the homogeneity of the elution profile and its shift leftward relative to MCR-1 in DDM (Supplementary Information).”

Please see below the nanodisc reconstitution experiment, performed using HPLC, which we have added as the first figure in the Supplementary Information. Nanodisc consisting of MSP1D1 and POPG lipid looked similarly promising compared to nanodisc consisting of MSP1E3D1 and POPG, but MSP1D1 is smaller in size, and so consistent with the relatively small size of MCR-1 (~65 kDa) we decided to move forward with MSP1D1 and POPG. *E. coli* Extract lipids (EcPolExt), which should more closely match the composition of the bacterial cell wall, were also tested, but we found nanodiscs comprised of POPG lipids to have more monodispersed peaks.

6. *Line 137 - Justification for picking MR6 - was it actually the best when you consider background in elisa for buffer/empty nanodisc? Clearly it worked in the end, but the whole reason for selection of that one over others isn't super clear*

We tested several Fabs during cryo-EM screening, including MR1 and MR5, but found a greater incidence of Fab dissociation and preferred orientation biasing in the data compared to data from sample incubated with the MR6 Fab. From the ELISA data in Extended Data Fig. 2b, c, we agree that the justification for choosing MR6 over the others is not very clear. However, the data nonetheless show MR6 to be a top binder, along with MR1 and MR5. To clarify, we have revised the line in question (now beginning at line 144 and ending at line 147) to state "... MR6 was identified as a top binder along with MR1 and MR5 (Extended Data Fig. 2b, c). However, MR6 was ultimately chosen based on two-dimensional (2D) class averages obtained from preliminary cryo-EM screening."

7. *Line 149 - ref Fig 1C here (rather than after a few sentences)*

We have revised the text to reference Fig. 1c as requested (now line 159) and removed its reference later in line 164.

8. *Line 154 - Can the authors confidently conclude zinc isn't present? When you fit the previous EPTA structure (5FGN) into the cryo map, there is density present at the expected Zn binding site, so not sure if a side chain has been modelled into that density instead? Do the authors have any biochemical evidence to suggest their purified protein doesn't bind zinc, or if this is just because they can't unambiguously assign it given the resolution?*

We thank the Reviewer for their question. Comparing the crystal structure of a di-Zn²⁺-bound MCR-1 periplasmic domain (PDB: 5LRM; from reference 28) to our full-length, substrate-bound cryo-EM structure, it can be seen that one Zn²⁺ - coordinated by Thr285 and Glu246 - is positioned directly above the phosphoethanolamine (PEtN) moiety belonging to the PE donor substrate in our cryo-EM structure (please see the attached figure below). In contrast, the other Zn²⁺ can be seen partially overlapping PE in our cryo-EM structure, leaving no space to model it in. From this comparison, the density in the region where the first Zn²⁺ is located is not at high enough resolution to distinguish between the side chain densities for Glu246 and Thr285 and density for Zn²⁺. Conversely, there is a clear lack of density in the region where the second Zn²⁺ is expected to reside. Given that this Zn²⁺ is expected to occupy a site so close to where the PE donor substrate binds in my cryo-EM structure, as well as the fact that it predicted to only be required for the transfer of PEtN to lipid A (and not cleavage of PEtN from PE) (please see reference 12), it is possible that this second Zn²⁺ is not recruited

until after PEtN has been cleaved from PE. This would explain the lack of density in our donor substrate-bound structure. Accordingly, we have revised the discussion section (lines 370-376) to better explain our observations.

Comparison of MCR-1 Zn²⁺-binding sites between di-zinc-bound periplasmic domain crystal structure and full-length substrate-bound cryo-EM structure. On left, the full-length cryo-EM structure of MCR-1 shown, depicted and colored as in Fig. 1. The di-zinc-bound crystal structure of the MCR-1 periplasmic domain (PD) (PDB: 5LRM) [28], shown in cartoon form and colored magenta, is superimposed onto the PD of the cryo-EM structure. Transparent cryo-EM density is shown overlaid the PD and PE substrate, colored green and yellow, respectively. On right, a zoomed in view of the two Zn²⁺-binding sites. From the crystal structure, the side chains of Glu246 and Thr285, which coordinate the first Zn²⁺ are shown in stick format and colored the same as the structure. Individual nitrogen, oxygen, and phosphorus atoms are colored blue, red, and orange, respectively. The two Zn²⁺ are shown as small purple spheres.

9. *Line 162-164 - sentence a little clunky, split into 2?*

We have revised the text to split the sentence in question into 2. The text now reads (beginning at line 175): “The upper portion of the PE density exhibits a shape that accommodates the PEtN headgroup (Fig. 2a, Extended Data Fig. 4). Situated nearby are a cluster of conserved residues, including the charged Glu116, Glu246, and Lys333, and polar His395, His466, His478, and the catalytic Thr285 (Fig. 2a).”

10. *Line 175 - refer to N108 and T112 having diminished activity and refer to numbers in Extended data fig 7b. Are these numbers from replicates? I think this needs to be made clear if any conclusions are being made, especially when these residues are particularly marginal*

We thank the Reviewer for their comment. All TLC data are representative of three biological replicates, which we have now stated in the Methods section (line 666). We would not feel comfortable averaging them all and doing a standard deviation unless we could run all samples on the same plate, which is not possible. We instead compare all the replicates and confirm they agree and show a representative gel.

11. *Line 198 - D102A and D102A/R190A mutants seem to be purified at lower levels than others, so perhaps has lower stability? Can the authors also justify why they did purifications rather*

than western blots to look at cellular accumulation, as all subsequent assays are done in the cell? Is the tagged version not functional?

We thank the Reviewer for their comment. By purifying the mutants, we show them to be properly folded and expressed at a similar level. For the purposes of the TLC assays, we utilize constitutive labeling with ³²P, so we are seeing cumulative lipid A modification over the entire growth. It is not a pulse labeling – we are not doing chromosomal expression, but overexpressing from a plasmid.

We have previously shown that even low levels of MCR-1/EptA expression from low-copy plasmids, like pWSK (1-5 copies per cell), results in robust PEtN modification (<https://doi.org/10.1128/jb.00067-23>; <https://doi.org/10.1111/mmi.13835>; and <https://doi.org/10.1128/JB.00498-21>). Minor differences in expression would therefore not result in major changes.

12. Line 212- Can the authors justify why they only tested some mutants in the polymixin assay and why those particular ones were chosen?

We thank the Reviewer for their question. The intent of the assay was to confirm the substrate binding pocket and that disrupting key residues impacts not only enzymatic activity, but antimicrobial resistance. To do this, we don't consider it necessary to test every residue. We took this approach for both the phospholipid binding site and the lipid A binding site. We strategically chose mutations that led to different degrees of lipid A modification. This was helpful, as it helped to demonstrate that you need a certain threshold of lipid A modification to maintain resistance.

13. Line 283 - Why are the non-lipid A species in the TLC? And why are they only in the 4' TLC?

We understand the Reviewer's curiosity concerning these lipid species. However, we feel confident they are not lipid A species. Depending upon the mutation to LPS structure, other phospholipids or phospholipid-linked glycans can flip to the cell surface. This leads to an increase in these lipids in our lipid A purifications. *E. coli* does make a number of minor lipids that remain to be characterized. If you made the authors guess, they would predict that they are a lipid-linked intermediate of ECA biogenesis. No matter, determining the nature of these lipid species is beyond the scope of the current study.

14. Throughout - authors alternate between single and three letter abbreviations for amino acids - would be good to pick one!

We thank the Reviewer for this comment and agree that this would be a good revision for the manuscript. Accordingly, we have revised the text and figures to use only single-letter abbreviations for amino acids.

Reviewer #3:

This is an extremely interesting and thorough structural investigation examining the mechanism of action of phosphoethanolamine transferase MCR-1 which catalyses the transfer of PEtN from a donor PE lipid bound to one site to modify Lipid A, increasing its overall charge. The authors use a combination of cryo-EM, biochemical mutagenesis, and molecular dynamics simulations to characterise a substrate-bound conformation of MCR-1. The cryoEM density maps show clear densities in two putative binding sites, one for the donor PE lipid, located ~20Å from the acceptor Lipid A site. The resolution was sufficient to allow both PE lipid and Kdo2-Lipid A to be unambiguously modelled into the second site. Mutations of key residues involved in the lipid A binding site showed a decrease in Lipid A modification, confirming the Lipid A binding site. Molecular dynamics simulations identified a major conformational change between the PE binding domain and the transmembrane domain, providing a novel mechanism for PEtN transfer to Lipid A.

This work is of significance across the fields of structural biology and microbiology, as it shows a previously unidentified mechanism for PEtN transferases that may be targeted for future antimicrobial design, and increases our general understanding of bacterial cellular processes.

The manuscript is clearly written and logically set out. The cryoEM, biochemical analysis and MD simulations supported each other. The results flowed and logically from one point to the next and supported the conclusions that were drawn in the manuscript. Overall, I had a few minor comments that should be addressed:

- 1. The authors describe State 1 and State 2 from the MD simulations, with State 1 corresponding to the cryoEM structure. They note that State 1 is consistent with the previously solved crystal structure of NmEptA identified by Anandan et al in 2017. They state that the NmEptA structure is apo, but in fact this is not strictly true, as there is a DDM detergent molecule bound in the PE binding site. Density corresponding to a Zn ion was also noted in the NmEptA structure.*

We thank the Reviewer for this observation. The text in lines 88-89 has been revised to clarify that the NmEptA crystal structure is not strictly in the apo state.

- 2. Anandan et al performed MD simulations on NmEptA and also noted a “tumble weed” type conformational change between the transmembrane domain and the catalytic domain. How did this second conformation captured by Anandan et al correspond to State 2 identified in the MD simulations presented here? From a quick inspection of the supporting information of Anandan et al, the two conformations appear quite different.*

We thank the Reviewer for this question. Indeed, Anandan *et al.* observed a significant conformational rearrangement in NmEptA, described as a “tumble weed” motion between the transmembrane and catalytic domains. While both our study and that of Anandan *et al.* report large-scale conformational changes, the specific orientations and nature of the structural transitions differ (please see the figure below). The State 2 conformation captured in our simulations represents a distinct configuration (panel a), likely stabilized by the interaction with KLA and the presence of a second Zn²⁺ ion, which were not considered in Anandan *et al.* simulations.

Interestingly, in one of our simulations of the apo-MCR-1 system, we observed a conformational rearrangement resembling that described by Anandan *et al.* for NmEptA (panel b), in which the periplasmic domain rolls over the membrane surface. However, in our simulations we did not observe the complete decoupling between the transmembrane and catalytic domains reported for NmEptA (panels a & b). In the simulation where this rearrangement occurs (Sim. 4), the salt bridge between D119 and K401, which we showed to be important for stabilizing the *state 1* conformation, is maintained. Moreover, a second salt bridge between D113 and K333 is also preserved throughout the trajectory (panels c & d).

Sequence comparison between *E. coli* MCR-1 and NmEptA reveals that the residue D113 in *E. coli* is substituted by a threonine in NmEptA, while D119 is replaced by an arginine (panel e). These differences suggest that the electrostatic interactions we observed in *E. coli* MCR-1 simulations, which appear to maintain the catalytic domain coupled to the transmembrane domain, are not conserved in NmEptA. This may facilitate the more extensive domain decoupling observed in NmEptA by Anandan *et al.*

Comparison between the MCR-1 and NmEptA MD simulations. **a**, Structural superposition of NmEptA (blue), from Anandan *et al.*, and State 2 (orange). **b**, Superposition of the MCR-1 cryo-EM structure (white), a conformation extracted from our simulations of apo-MCR1 (green) and NmEptA (blue). The conformation from our MD simulation exhibits a similar “tumble weed” motion as observed for NmEptA but retains coupling between the periplasmic and transmembrane domains. **c**, Zoomed-in view of the conformation from our MD simulations showing the salt-bridges that prevent the complete dissociation between the periplasmic and transmembrane domains. **d**, Frequency of the salt-bridges formation between the periplasmic and transmembrane domains during the simulations. **e**, Sequence alignment between *E. coli* MCR-1 and NmEptA indicating that key electrostatic interactions stabilizing domain coupling in *E. coli* are not conserved in NmEptA.

- In terms of the MD simulations, it would be good to know which residues coordinated the Zn ions, and how stably Zn bound to the binding site of the protein. For what proportion of the simulation did Zn remain stably coordinated by the proposed Zn binding residues? Both the first and second Zn ions were placed in the respective binding sites based on crystallographic information from other proteins.*

We thank the Reviewer for this question. The first Zn²⁺ ion was placed using the crystal structure of *Neisseria meningitidis* EptA (NmEptA), which served as a reference for our State 1 conformation. In the initial model, the Zn²⁺ ion is coordinated by Glu246, Thr285, His466,

and Asp465, consistent with the coordination observed in the crystal structure. During the MD simulations, this ion remained stably bound, with Glu246 and Asp465 maintaining strong coordination throughout the trajectory, while coordination by His466 and Thr285 was more transient.

The second Zn^{2+} ion was modelled based on the crystal structure of the soluble domain of MCR-1 (PDB ID: 5LRM), where it is initially coordinated by Tyr285 (PEtN-Thr285 in our simulations), His395, and His478. Over the course of the simulation, we observed a slight repositioning of this ion, accompanied by a loss of interaction with the histidine residues. Instead, it became primarily coordinated by Tyr285 and a nearby KLA phosphate group. This repositioning appears to favour interaction with the substrate and may facilitate catalysis.

To clarify these observations, we have added a new extended data figure (Extended Data Fig. 10) illustrating the Zn^{2+} coordination throughout the simulations.

Extended Data Fig. 10. Zinc coordination. **a**, A zoomed in panel depicting the Zn^{2+} ion binding site in the State 1 conformation. The ion was modelled based on the *NmEptA* structure (PDB ID: 5FGN) and is coordinated by the residues E246, T285, D465 and H466. **b**, Time traces of the distance between Zn^{2+} and the coordinating residues E246 (top left), T285 (top

right), H466 (bottom left) and D465 (bottom right). **c**, A zoomed in view showing the ions Zn^{2+} binding sites in the State 2 conformation. The ions coordinates were modelled using the di-zinc MCR-1 structure (PDB ID: 5LRM). The first Zn^{2+} ion is coordinated by the same residues as in State 1. The second ion is coordinated by the residues T285, H395 and H478. **d**, Time traces of the distance between the first Zn^{2+} ion and the residues E246 (top left), T285 (top right), H466 (bottom left) and D465 (bottom right). **e**, Final frame of one of the MD simulation repeats showing the repositioning of the second Zn^{2+} ion and its interaction with KLA. **f**, Time traces of the distance between the second Zn^{2+} ion and the residues T285 (top left), H395 (top right), H478 (bottom left) and KLA (bottom right). In our simulations, the second Zn^{2+} ion is primarily coordinated by T285 and a nearby KLA phosphate group.

- The methods describe an unusual set-up for the simulations. If I am understanding correctly, the authors first coarse-grained the structure and used the Martini 3 force field, applied an elastic network (with a high force constant) to the backbone, built a CG PE/PG membrane around the protein, ran 1 μs of CG simulations, then backmapped it to atomistic resolution, then ran replicate 500 ns atomistic simulations under different conditions and analysed these. This seems like an awfully convoluted way to set up a simulation. Why was this done? Why not just take the cryoEM structure and equilibrate it in an atomistic membrane simulations? What purpose do the CG simulations serve, except to cite a specific set of tools for CG membrane generation and back mapping?*

We thank the reviewer for this comment. The coarse-grained (CG) simulation step, following the MemProtMD pipeline (Stansfeld *et al.*, 2015), was employed to embed the protein within a lipid bilayer in an automated and unbiased manner. This strategy enables the construction of the membrane around the protein, followed by 1 μs CG simulation to equilibrate the membrane–protein system. The CG simulations are used rather than atomistic as this takes a shorter period to permit thorough lipid mixing and equilibration. The resulting structure is subsequently converted to atomistic resolution with CG2AT for atomic-level MD simulations; with the coordinates from the cryo-EM structure used for the protein. This method has been widely adopted in the membrane protein field as it avoids manual placement of lipids and provides a reproducible and automated way to generate membrane-embedded systems.

- There are also gaps in the computational methodology that prevent reproducibility. For example, which Martini water was used? I'm assuming Martini 3 water parameters, but this should be explicitly stated. The RMSD between the back mapped and cryo-EM structure should also be given. I would expect this to be very low given the high force constant that has been used in the elastic network.*

We thank the Reviewer for this comment. We have now clarified in the Methods section (lines 668-698) that the Martini 3 polarizable water model was used in the coarse-grained simulations.

Regarding the back mapping, we employed the *align* protocol in CG2AT2 (Vickery *et al.*, 2021), which fits the original atomistic structure (derived from the cryo-EM model) to the coarse-grained coordinates. As CG2AT2 aligns the input atomistic structure to the CG backbone, the resulting back mapped structure remains virtually identical to the cryo-EM model, with minimal deviations arising from the fitting process. The backbone RMSD after alignment is very low (0.137Å), confirming that the cryo-EM conformation is retained.

- Restraints were used in the equilibration period of the atomistic simulations. What was the relaxation protocol for the restraints? How was the system equilibrated?*

We thank the Reviewer for this question. After conversion to atomistic resolution with CG2AT, the system was equilibrated for 10 ns under NPT conditions with positional restraints on the protein non-hydrogen atoms. The MemProtMD pipeline was used not only to insert the

protein into the membrane in an unbiased and automated manner, but also to pre-equilibrate the membrane protein system. In this protocol, the membrane is allowed to equilibrate around the protein in CG resolution. As a result, our system was already equilibrated prior to atomistic simulation, reducing the need for extensive relaxation protocols. Additionally, the CG2AT back mapping procedure includes an initial NVT equilibration step following the atomistic conversion.

- 7. Molecular dynamics checklist point 4c asks. "Are other parameters for the system setup described in the text, such as protonation state, type of structural restraints if applied, nonbonded cutoff, thermostat and barostat, etc.?" These have not been included in the manuscript. Please explicitly state what force field, water model and ion model was used for the atomistic simulations, and how the protonation state was assigned. There is no reference to the atomistic force field in either the methods or in Extended Data Table 5,. Likewise, there is no reference to the pressure coupling or temperature coupling TauP or TauT, or to the cut-off method for the nonbonded interactions. Without this information, the simulation protocol cannot be reproduced.*

We thank the Reviewer for pointing out the missing details. The Methods section (lines 668-698) has been updated to include all relevant parameters for the atomistic simulations. These include the CHARMM36m force field, TIP3P water model, treatment of protonation states, thermostat and barostat settings, coupling constants, and nonbonded interaction cut-offs.

Response to reviewers

Zinkle et al. “Mechanistic basis of antimicrobial resistance by the phosphoethanolamine transferase MCR-1”

We thank the Reviewers again for their helpful comments, critical assessment, and positive appraisal of our manuscript. Below is our point-by-point response to the most recent comments from Reviewers #1, #2, and #3:

Reviewer #1:

The authors have significantly revised the manuscript and addressed all the major concerns. They have modified some figures and have responded to most of the reviewers' suggestions appropriately. Their responses are convincing. I have one minor suggestion and one comment:

We thank the Reviewer for their positive assessment of our revised manuscript.

Minor suggestion:

Extended Data Fig. 3 is improved but still not very clear. I had to read it multiple times to fully understand what is shown. In the rebuttal, the authors explained: “we did not re-use these particles, but rather used the new volume (3.73 Å) as an input volume for a new round of heterogeneous refinement using the original 383,413 particle stack and replacing the original input volume representing MCR-1 in nanodisc with Fab (shown in green).” This should also be mentioned in the figure legend, with the new 3.73 Å volume marked as the initial model for the new round of heterogeneous refinement. While this is clearly described in the Methods section, adding it to the figure legend and marking it in the figure would improve clarity.

We thank the Reviewer for this suggestion, which we agree will improve clarity. Accordingly, we have revised the figure legend and marked the 3.73 Å map in the extended figure.

Comment:

Regarding Reviewer #1, Question 6: the authors provided an additional set of eight 2D class averages showing the transmembrane domain (circled in red). However, the TM region is still not very clear. Many cryo-EM papers on GPCRs, transporters, and efflux pumps published over the last 5–10 years show transmembrane regions in detergent, liposomes, or lipids more clearly than in the present case. That said, this does not affect the 3D reconstruction, the overall findings, or the conclusions of the manuscript.

We agree that other 2D classifications on small membrane proteins have clearer views of their respective TM regions. Unfortunately, we cannot say with absolute certainty why the TM region in our 2D classes is not similarly clear (we can speculate, for example, on how the flexibility of the nanodisc and/or periplasmic domain of MCR-1 might contribute to a more blurry appearance in the TM domain). Nonetheless, as the Reviewer acknowledged, the reconstruction is unaffected, and the quality of the density is high.

I have no further questions or comments. This manuscript is suitable for publication.

We thank the Reviewer again for their supportive comments.

Reviewer #2:

The manuscript remains timely and novel. My positive comments about the manuscript from the first round of review remain: "This manuscript reveals a cryo-EM structure, functional and computational data to improve our understanding of the MCR-1 mechanism. MCR-1 is an enzyme directly responsible for polymyxin resistance, by adding PEtN group onto lipid A, the target of polymyxin. Overall, this manuscript is clear, timely, and contributes important knowledge to the field on how

MCR-1 (and similar enzymes) function. It combines really multi-disciplinary and complementary techniques to present a convincing two step model of MCR-1 mechanism. The figures and text are clear throughout."

The below comments are specifically in response to the authors response:

The authors seemed reluctant to partake in any of the minor lab-based experiments suggested for this paper. The justification for not doing western blots was fine, but still it seems like a fairly quick and simple experiment to do that would have been insightful. Furthermore, the justification for selecting only some of the mutants in their experiments was not that clear, and in some cases not explained, when again it should have been fairly simple to add on.

I don't think doing these additional experiments are essential for the paper to be published, but it would have been an improvement for fairly little hands-on time.

All of the easier, text-based, edits were addressed well.

We thank the Reviewer again for their positive assessment of our revised manuscript. With respect to our justification for selecting only some of the mutants in our polymyxin resistance assay, we thought it was unnecessary to test every residue-of-interest in the PE and KLA binding sites, but rather just the mutations that led to different degrees of lipid A modification. This could then help to 1) further validate the substrate binding sites; and 2) show that disrupting key residues not only impacts enzymatic activity, but also susceptibility of the enzyme to polymyxins.

Reviewer #3:

The authors have adequately addressed my queries in their revised manuscript.

We thank the Reviewer for their support of our manuscript.